# Population analysis of *Legionella pneumophila* reveals a basis for resistance to complement-mediated killing

Bryan A. Wee [1,9], Joana Alves [1,9], Diane S. J. Lindsay [2], Ann-Brit Klatt[3], Fiona A. Sargison [1], Ross L. Cameron[4], Amy Pickering [1], Jamie Gorzynski [1], Jukka Corander[5,6], Pekka Marttinen [7], Bastian Opitz [3], Andrew J. Smith [2,8] & J. Ross Fitzgerald [1✉]

*Legionella pneumophila* is the most common cause of the severe respiratory infection known as Legionnaires' disease. However, the microorganism is typically a symbiont of free-living amoeba, and our understanding of the bacterial factors that determine human pathogenicity is limited. Here we carried out a population genomic study of 902 *L. pneumophila* isolates from human clinical and environmental samples to examine their genetic diversity, global distribution and the basis for human pathogenicity. We find that the capacity for human disease is representative of the breadth of species diversity although some clones are more commonly associated with clinical infections. We identified a single gene (*lag-1*) to be most strongly associated with clinical isolates. *lag-1*, which encodes an *O*-acetyltransferase for lipopolysaccharide modification, has been distributed horizontally across all major phylogenetic clades of *L. pneumophila* by frequent recent recombination events. The gene confers resistance to complement-mediated killing in human serum by inhibiting deposition of classical pathway molecules on the bacterial surface. Furthermore, acquisition of *lag-1* inhibits complement-dependent phagocytosis by human neutrophils, and promoted survival in a mouse model of pulmonary legionellosis. Thus, our results reveal *L. pneumophila* genetic traits linked to disease and provide a molecular basis for resistance to complement-mediated killing.

---

[1] The Roslin Institute, Royal (Dick) School of Veterinary Studies, University of Edinburgh, Edinburgh, Scotland, UK. [2] Bacterial Respiratory Infections Service (Ex Mycobacteria), Scottish Microbiology Reference Laboratory, Glasgow, Scotland, UK. [3] Department of Internal Medicine/Infectious Diseases and Pulmonary Medicine, Charité Universitätsmedizin Berlin, Berlin, Germany. [4] NHS National Services Scotland, Health Protection Scotland, Glasgow, Scotland, UK. [5] Department of Mathematics and Statistics, University of Helsinki, Helsinki, Finland. [6] Department of Biostatistics, University of Oslo, Oslo, Norway. [7] Helsinki Institute for Information Technology, Department of Computer Science, Aalto University, Aalto, Finland. [8] College of Medical, Veterinary & Life Sciences, Glasgow Dental Hospital & School, University of Glasgow, Glasgow, UK. [9]These authors contributed equally: Bryan A. Wee, Joana Alves. ✉email: Ross.Fitzgerald@ed.ac.uk

*L*egionella pneumophila is a γ-proteobacterial species that parasitises free-living amoeba in freshwater environments[1,2]. *L. pneumophila* hijacks the phagocytic process in amoebae and human alveolar macrophages by subverting host cellular mechanisms to promote intracellular replication[3]. *Legionella* infections are a global public health concern presenting as either a severe pneumonia known as Legionnaires' disease or Pontiac fever, a self-limiting flu-like syndrome[4–7]. Importantly, recent surveillance studies have indicated a steady increase of legionellosis incidences globally[8,9].

*L. pneumophila* infection in humans is considered to be the result of accidental environmental exposure and the selection for pathogenic traits among *L. pneumophila* is likely to be driven by co-selective pressures that exist in its natural habitat[10]. The pivotal mechanism required for intracellular replication is the type IV secretion system (T4SS) that is conserved across all known members of the genus *Legionella*[11,12]. A very large repertoire of effector proteins in different combinations is encoded by *Legionella* species and can be secreted by this system to mediate critical host-pathogen interactions[11]. In addition, the ability to infect eukaryotic cells has evolved independently many times[11]. Despite this, less than half of all currently described *Legionella* species have been reported to cause human disease[13]. Furthermore, there is an over-representation of a single serogroup (Sg-1) of *L. pneumophila* in human infections, which is responsible for more than 85% of all reported cases of legionellosis[14]. Sg-1 strains can be further subdivided phenotypically using monoclonal antibodies (mAbs) that recognise various components of the lipopolysaccharide (LPS). The most prevalent mAb subtype in human infections is associated with an LPS *O*-acetyltransferase enzyme encoded by the *lag-1* gene (*lpg0777*), which confers an LPS epitope recognised by the mAb 3/1 from the Dresden mAb panel[15,16]. *lag-1* has previously been reported to be associated with clinical isolates of *L. pneumophila* but its role in pathogenicity remains a mystery[17–21].

Recently, it was shown that a very limited number ($n = 5$) of *L. pneumophila* sequence types (ST)s are responsible for almost half of all human infections and that these clones have undergone very recent emergence and expansion[22]. In addition, it was shown that recombination between the dominant STs has led to sharing of alleles that may be beneficial for human pathogenicity[22]. However, the genetic basis for the enhanced human pathogenic potential of these STs is unknown. Early studies identified the capacity for some *L. pneumophila* strains to resist killing in human serum, a phenotype that may correlate with increased virulence[23]. However, the molecular mechanism of strain-dependent serum resistance by *L. pneumophila* has remained elusive for over 30 years.

Here, we employ a population genomic analysis of *L. pneumophila* isolates from clinical and environmental sources to investigate the diversity of genotypes associated with human disease. Further, we employ genome-wide association analyses to identify genetic traits associated with human pathogenic strains revealing *lag-1* to be the most strongly associated determinant of human pathogenic potential in *L. pneumophila*. We demonstrate that acquisition of *lag-1* has occurred widely across the species by recent horizontal gene transfer and recombination, leading to strains that have enhanced resistance to complement-mediated killing, neutrophil phagocytosis and survival in a murine model of pneumonia.

## Results and discussion
### The potential for human infection is distributed across the *L. pneumophila* species phylogeny. In order to examine the diversity of *L. pneumophila* associated with human clinical

infection in comparison to those from environmental sources, we carried out whole-genome sequencing of 397 clinical and environmental *L. pneumophila* subsp. *pneumophila* isolates from an archived collection (1984 to 2015) of *Legionella* species held at the Scottish Haemophilus, Legionella, Meningococcus & Pneumococcus Reference Laboratory, Scottish Microbiology Reference Laboratory, Glasgow (SHLMPRL) as detailed in the Methods section. The dataset included 166 clinical isolates from infected patients, primarily from sputum and bronchoalveolar lavages (BAL). The proportion of clinical isolates from the less severe Pontiac fever form of legionellosis is unknown but is presumed to be very limited due to low hospitalisation rates for these infections. In addition, we included sequences from 231 environmental isolates of *L. pneumophila* subsp. *pneumophila* from water sources such as cooling towers and plumbing systems sent to the reference laboratory for routine testing and surveillance (Supplementary Data 1). Some of the isolates classified as environmental in our dataset were sampled as part of an outbreak investigation and therefore may in some cases be related to clinical isolates. Accordingly, the power to identify traits that differentiate clinical and environmental groups may be negatively affected. To place the SHLMPRL isolates into context of the known global diversity of *L. pneumophila* subsp. *pneumophila* isolates, we included 500 assembled whole genomes that were available in the public database (NCBI Genbank). We constructed a Maximum-Likelihood phylogenetic tree from 139,142 core genome single nucleotide polymorphisms (SNP), which indicates the segregation of the *L. pneumophila* subsp. *pneumophila* population into seven major clades (Fig. 1) each supported by a minimum of one and a maximum of two sub-clusters defined by Bayesian Analysis of Population Structure (BAPS) analysis (Supplementary Fig. 1). Isolates from the SHLMPRL collection were distributed among all major clades indicating that the isolates are representative of the global species diversity (Fig. 1). Although recombination has played an important role in diversification of the species, complicating the accurate reconstruction of the phylogeny[24], the major phylogenetic clusters are consistent with those identified in previous population studies employing fewer isolates[25,26]. Recombination is more likely between phylogenetically related (and genetically similar) isolates, which may amplify the phylogenetic signal that defines these major groups[27]. Of the 166 clinical isolates, 80 belonged to the five most common sequence types (ST) implicated in human infections (ST1, 23, 36/ 37, 47 and 62). The remaining 86 clinical isolates were from STs distributed across all seven major phylogenetic clusters (Fig. 1). These data indicate that although only 5 STs are responsible for almost half of all infections, the other half of human infection isolates come from diverse genetic backgrounds distributed across the species. Of note, five clusters from distinct outbreaks of Legionnaires' disease in Scotland between 1985 and 2012[28–30] originated from different clades (Fig. 1). Similarly, the 231 environmental isolates are also distributed across all major phylogenetic groups (Fig. 1). Notably, 35% (81 of 231) of environmental isolates belong to one of the five clinically dominant STs suggesting that at least a third of all environmental *L. pneumophila* from the sampled sources in Scotland have human pathogenic potential. Overall, although some STs appear to have higher human pathogenicity, our findings support the understanding that the capacity for causing human disease is widely distributed across diverse *L. pneumophila* genetic backgrounds.

### Genome-wide association analysis of *L. pneumophila* reveals genetic traits associated with human pathogenicity. The factors that contribute to the enhanced human infectivity or transmissibility of some *L. pneumophila* clones are unknown. Our

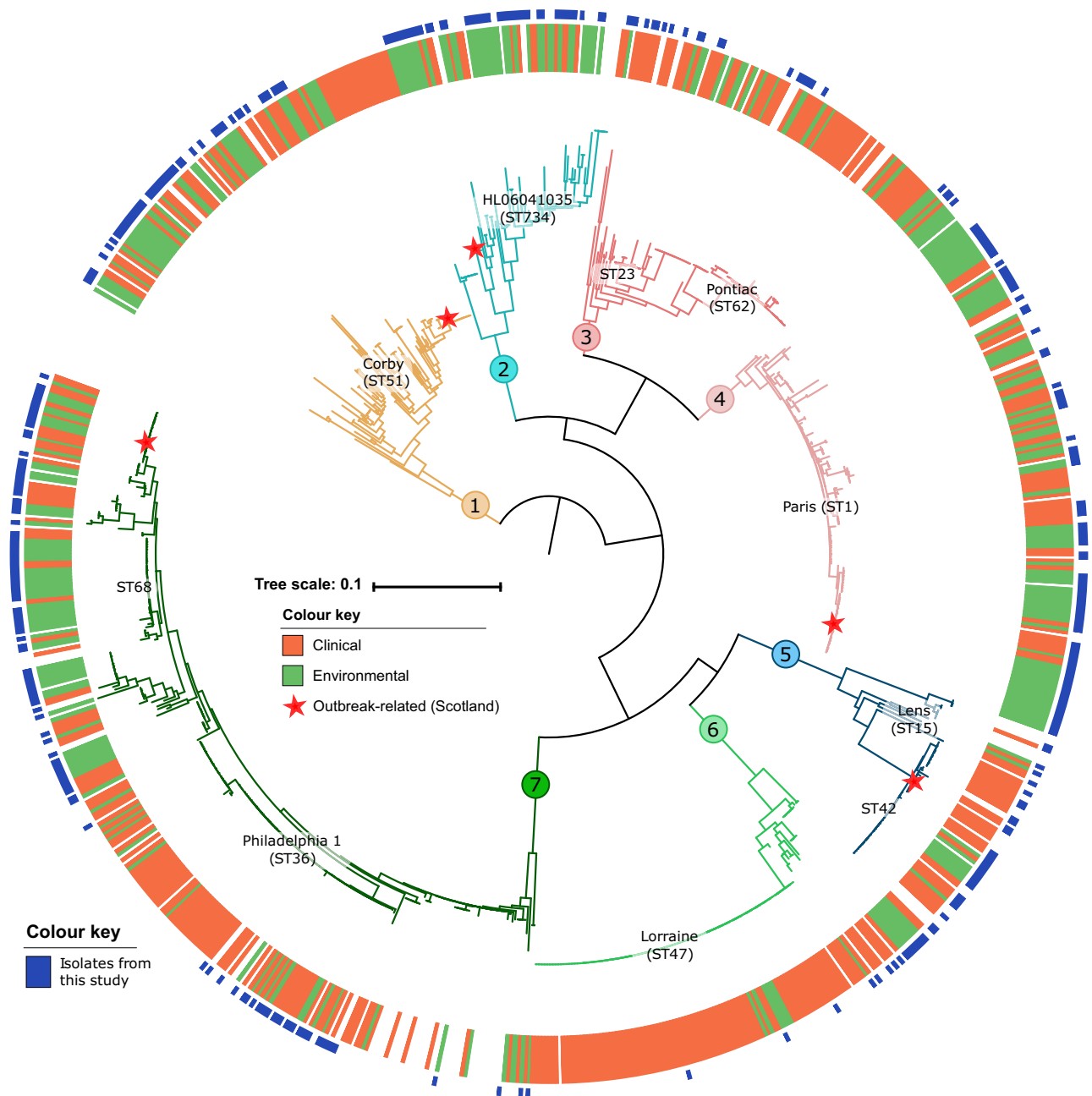

**Fig. 1 Human clinical isolates of *L. pneumophila* are widely distributed across the global species diversity.** A maximum-likelihood phylogeny of 902 isolates based on 139,142 core genome SNPs divides the subspecies into seven major clades that are also supported by BAPS clustering. Isolates linked to five major outbreaks that have occurred in Scotland, UK between 1985 and 2012 originated from different lineages are indicated. In addition, isolates associated with sporadic cases of human infection have also emerged from diverse genetic backgrounds across all major clades. For reference, the position of five major disease associated clones (ST1, 23, 36/37, 47 and 62) and well-characterised reference genomes are indicated. The tree scale indicates the number of subtitutions per site.

strategy, to address this gap in understanding, involved sequencing of large numbers of genetically diverse environmental isolates, affording for the first time, a large-scale genome-wide association study (GWAS) of clinical vs environmental isolates to explore the bacterial genetic basis for human clinical disease. Initially, we performed systematic subsampling on our dataset to reduce the number of closely related isolates originating from the same type of source, either clinical or environmental, while retaining genetic diversity. This step removes over-represented endemic or epidemic clonal lineages in the dataset that arise from opportunistic, convenience sampling and also represents an additional control for a stratified bacterial population structure. From this reduced dataset ($n = 452$), we used the programme SEER to identify a total of 1737 $k$-mers that were enriched (significantly associated, $p < 0.05$) among clinical isolates in comparison to environmental isolates. Mapping to the *L. pneumophila* Philadelphia 1 Sg-1 reference genome (Accession number: AE017354) [https://www.ncbi.nlm.nih.gov/nuccore/AE017354.1] revealed that 39% ($n = 673$) of the $k$-mers aligned to a region of the genome spanning between loci *lpg0748* and *lpg0781* representing an 18 kb cluster of genes involved in LPS biosynthesis and modification (Fig. 2)[31]. A total of 22 genes in this cluster each had at least one

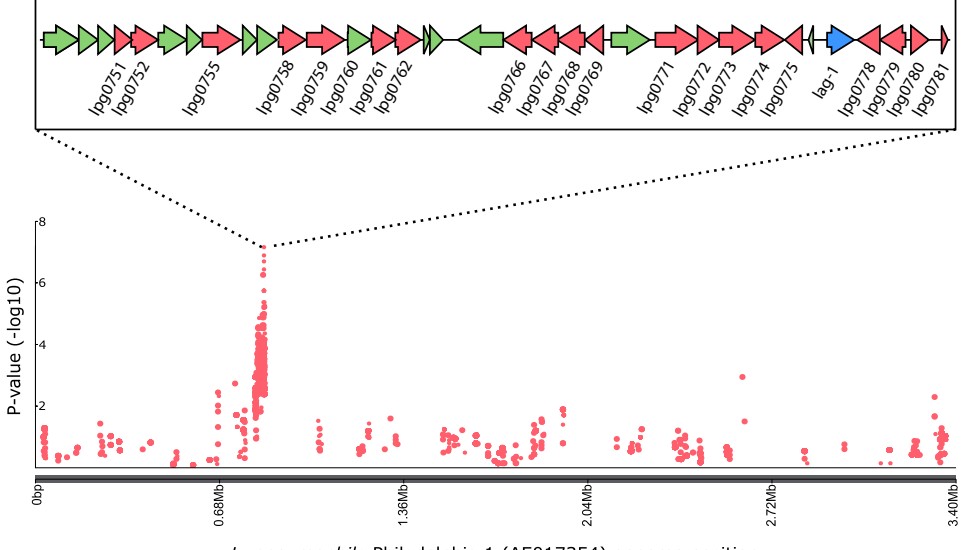

**Fig. 2 GWAS reveals gene sequences associated with clinical isolates of _L. pneumophila_.** Manhattan plot showing the genomic position of _k_-mers significantly associated with clinical isolates. The LPS biosynthesis and modification locus (822,150bp–855,010 bp: _lpg0751-lpg0781_) in the Philadelphia 1 reference genome (AE017354) [https://www.ncbi.nlm.nih.gov/nuccore/AE017354.1] is represented and genes with significant _k_-mer associations coloured in red, with _lag-1_ in blue.

significantly enriched _k_-mer that mapped to it: namely, _lpg0751, lpg0752, lpg0755, lpg0758, lpg0759, lpg0760, lpg0761, lpg0762, lpg0766, lpg0767, lpg0768, lpg0769, lpg0771, lpg0772, lpg0773, lpg0774, lpg0775, lpg0777 (lag-1), lpg0778, lpg0779, lpg0780_ and _lpg0781_ (Fig. 2 and Supplementary Table 1). Two additional conservative SEER analyses using more stringently subsampled datasets resulted in a smaller number of significantly enriched _k_-mers (205 and 61, respectively), that converge on a single gene within the LPS cluster (_lpg0777, lag-1_) that encodes an _O_-acetyltransferase involved in modification of the O-antigen of the _L. pneumophila_ Sg-1 LPS (Supplementary Fig. 2). To account for the random nature of the subsampling algorithm, multiple iterations were performed at the middle threshold, and the distribution of significant k-mers was shown to be consistent across all (Supplementary Fig. 2b).

To corroborate the initial findings of the _k_-mer based SEER approach, we employed a different GWAS method (SCOARY) that examined the distribution of orthologous genes using the pan-genome pipeline ROARY[32,33]. Consistent with SEER, this approach also indicated that the Sg-1 LPS biosynthesis genes were enriched among clinical isolates with _lag-1_ exhibiting the strongest statistical support over other Sg-1-associated LPS genes (Supplementary Table 1). The corrected (Benjamini–Hochberg) _p_-value for _lag-1_ is several orders of magnitude lower (9.74E-11) than other Sg-1 LPS genes that were above the significance threshold (_lpg0779_: 2.35E-05, _lpg0780_: 2.35E-05, _lpl0815/lpg0774_: 3.84E-05 and _lpg0767_: 6.92E-05). As mentioned, _lag-1_ has previously been reported to be prevalent among clinical isolates[17–21]. However, our large objective pangenome-wide analysis of the _L. pneumophila_ species indicates that of all 11198 accessory genes, _lag-1_ has the strongest association with clinical isolates, suggesting a major role in human disease. In total, 80.1% of clinical isolates contained _lag-1_ compared to 30.8% of environmental isolates (Supplementary Fig. 3). Sg-1 LPS genes were found in high-frequency throughout the population, and in each major lineage indicating species-wide gene transfer. The Sg-1 LPS cluster is also found in other subspecies of _L. pneumophila_ that can infect humans, such as subspecies _fraseri_ and _pascullei_ but has not been reported in other species of _Legionella_[34,35]. Here, we observed that _lag-1_ can be associated with different combinations of Sg-1 LPS genes and still express the expected

_lag-1_ phenotype represented by the mAb 3/1-specific epitope (Dresden mAb scheme) (Supplementary Data 1)[36,37]. However, genetic instability and phase variation has been reported to affect the mAb phenotype between closely related strains[38,39]. One mechanism of phase variation is the excision of a 30 kb genetic element that results in a change in mAb specificity manifested by a loss of virulence in guinea pigs and loss of resistance to complement[40,41].

**Recombination has mediated the dissemination of three dominant _lag-1_ alleles across the _L. pneumophila_ species.** It has been reported since the 1980s that isolates expressing the epitope recognised by the mAb 3/1 are more frequently associated with isolates from community-acquired and travel-associated infections[37,42,43]. However, the pathogenic basis for this association remains a mystery. In order to further investigate the role of the clinically-associated gene _lag-1_, we examined its diversity and distribution across the 902 _L. pneumophila_ genomes employed in the current study. In total, three major allelic variants of _lag-1_ that had been previously identified to be representative of reference strains, Philadelphia, Arizona, and Corby, respectively, were identified[43]. In our dataset, variant 1 (Philadelphia), was present in 195 (22%), variant 2 (Arizona) in 195 (22%) and variant 3 (Corby) in 178 (20%) of the 902 isolates examined (Fig. 3a). Each variant was distributed across the phylogeny, with variant 1 found in 3 major clades (1, 4 and 7) and variants 2 and 3 identified among isolates of all 7 major clades (Fig. 3a). Of note, clade 2 has the lowest frequency (12%) of isolates encoding the _lag-1_ gene and is also characterised by an under-representation of clinical isolates (31%). In addition to the three major _lag-1_ alleles, we identified a relatively small number (n = 40) of derived minor allelic variants that differ by <1% nucleotide identity from any of the 3 major variants (Supplementary Table 2). Of these, only 10 are predicted to encode for full-length proteins suggesting most are likely to be non-functional pseudogenes. The occurrence of these three allelic variants of _lag-1_ gene and their widespread distribution across the species phylogeny indicates frequent horizontal dissemination of recently acquired _lag-1_ alleles driven by a strong selection

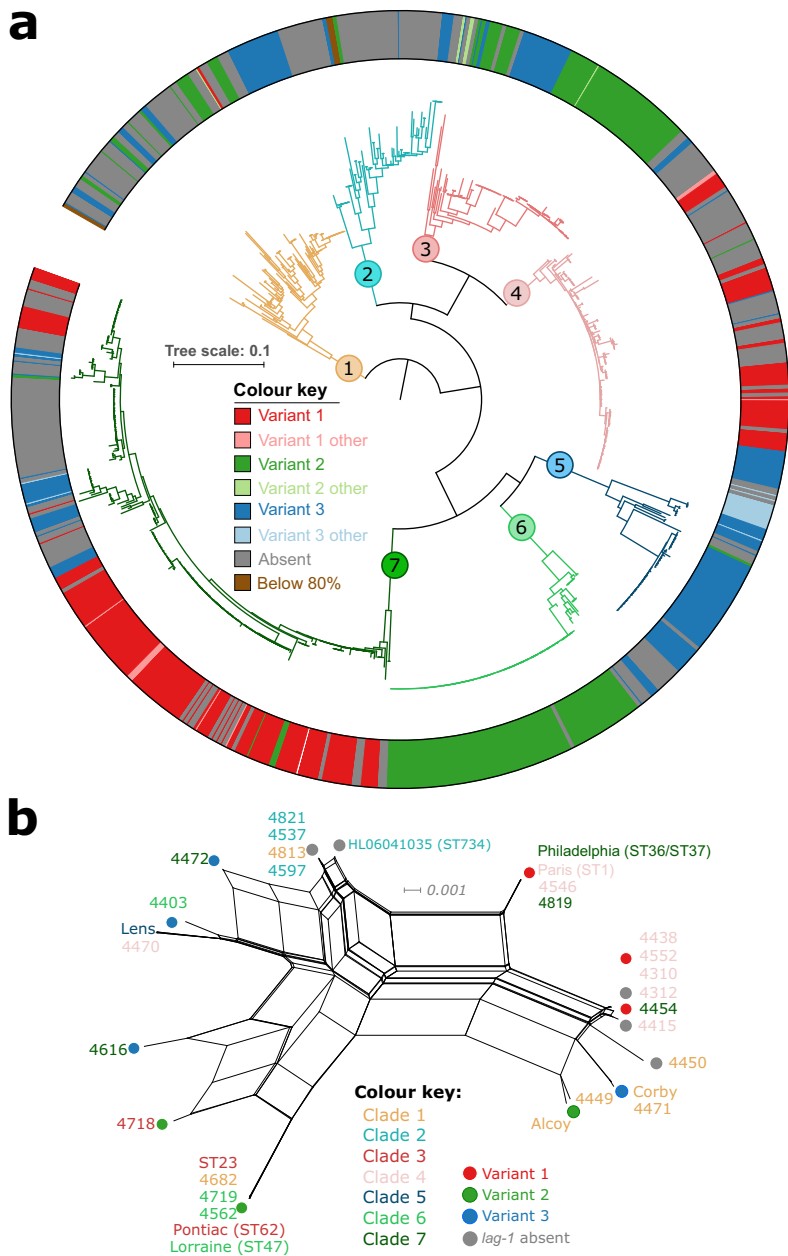

**Fig. 3 lag-1 variants have disseminated horizontally across the species. a** Core-genome SNP-based phylogenetic tree indicating the distribution of the 3 major allelic variants of *lag-1* across Sg-1 strains. Variant 1 (Philadelphia, Red), Variant 2 (Arizona, Green) and Variant 3 (Corby, Blue). Each major clade is associated with at least two different *lag-1* alleles. Minor alleles that are within >99% nucleotide sequence identity to a major variant are shown with lighter colours. Isolates not containing *lag-1* are coloured grey. The tree scale indicates the number of subtitutions per site. **b** A neighbour-net phylogenetic network indicating recombination of Sg-1 LPS genes including *lag-1*. The network was drawn using uncorrected P-distances with the equal angle method in Splitstree from a concatenated alignment of 10 conserved Sg-1 LPS genes that were significantly associated with clinical isolates (9405 positions). Orthologs of 10 LPS cluster genes were extracted from 16 phylogenetically representative isolates coding 3 different variants of *lag-1*, 7 isolates missing *lag-1* and 9 reference genomes (Lens, HL06041035, Philadelphia-1 [ST36], Paris [ST1], Corby, Alcoy, Pontiac [ST62], Lorraine [ST47], ST23).

pressure for the conservation of *lag-1* sequence and function. The nucleotide sequence identity of 89%–95% between the three allelic variants is lower than the average nucleotide sequence identity for genes across the species, which is typically >95% (Supplementary Table 2). The closest homologue of *lag-1* in the NCBI non-redundant protein database shares only 45% amino acid sequence identity and encodes a putative acyltransferase present in an environmental species of the genus *Pseudomonas*. The selective advantage for *L. pneumophila* to maintain *lag-1* in the environment is still unclear. It has been proposed that the

increased hydrophobicity of LPS when acetylated by *O*-acetyl-transferase may enhance *L. pneumophila* survival in amoebae vacuolar compartments[44]. An alternative hypothesis is that *L. pneumophila* is evolving the capacity to transmit between humans or from humans to the environment as proposed by David et al.[22]. In which case, the recent spread of *lag-1* across the phylogeny may have been driven by selection in the human host and contributed to the recent expansion of successful clones. Horizontal transfer and recombination events involving the 18 kb LPS biosynthesis gene cluster has been reported previously,

including between Philadelphia-1 (ST36) and Paris (ST1)[45,46]. In addition, a microarray-based comparative genomic study identified this genomic region to be associated with human disease[47]. To investigate the potential role of recombination in the distribution of the LPS genes including *lag-1*, we carried out a split network analysis based on the concatenated alignment of 10 LPS biosynthesis genes that were present in 23 representative isolates from across the phylogeny encoding different *lag-1* variants (Fig. 3b). This analysis revealed extensive reticulation consistent with recombination across the gene cluster and identified horizontal transfer of LPS genes between three major clinical sequence types, ST23, ST47 and ST62, respectively (Fig. 3b). This network analysis revealed that even highly similar LPS biosynthesis gene clusters, such as those found in isolates 4454, Corby and Alcoy or 4616 and 4718, encode different variants of *lag-1*, consistent with recent gene conversion of *lag-1*. We also identified three examples of closely related genomes (average nucleotide identity across the genome of >99.8%) that exhibited a signature of homologous recombination affecting *lag-1* and the surrounding genomic region (Supplementary Fig. 4). These findings expand on a previously proposed mechanism of *lag-1* gene deletion by Kozak and colleagues[43] and demonstrate that recombination can restore a disrupted or missing *lag-1* gene or replace one functional *lag-1* variant with another (Supplementary Fig. 4).

A comparison of re-assortment rates across major phylogenetic lineages showed no significant differences between LPS and non-LPS genes across the *L. pneumophila* genome ($p = 0.6275$). This suggests that recombination is active on a genome-wide scale, in agreement with earlier analyses of rates of recombination across the genome[45,46] (Supplementary Fig. 5 and Supplementary Data 2). Taken together, these findings highlight the wide dissemination of *lag-1* and other LPS genes by recombination.

**lag-1 confers resistance to killing by human plasma.** Our population data indicate that *lag-1* gene has the highest statistical support of all accessory genes for a role in human pathogenicity. Consistent with this, previous studies have reported an epidemiological correlation between *lag-1* or its mAb 3/1-recognised epitope and clinical disease[17–21] but a mechanistic understanding of its role in pathogenesis has proved elusive. Of note, an early study described an increased ability of an endemic nosocomial *L. pneumophila* strain to survive complement killing when compared to an environmental strain collected from the same medical facility[23]. More recently, resistance to complement-mediated killing has been proposed as a key fitness advantage that enables the Philadelphia Sg-1 strain to establish infections compared to other serogroups[48]. To investigate this phenotype in the context of *lag-1* genotype, we examined resistance to killing in human plasma for the aforementioned 23 Sg-1 *L. pneumophila* strains selected to represent the diversity of *lag-1* genotypes from across the phylogeny. We observed that all strains lacking a *lag-1* gene were susceptible to killing in plasma, whereas groups of isolates containing *lag-1 variants* exhibited enhanced resistance, independent of the *lag-1* variant encoded (Fig. 4a).

Within our dataset, we identified a group of epidemiologically-related isolates from a single Scottish healthcare facility typed as ST5 (a single locus variant of ST1), which were predicted to vary with regard to *lag-1* functionality. ST5 has only been found in this location, to date. The earliest isolates obtained from a nosocomial outbreak in 1984/1985[30] contained variant 3 (Corby) of the *lag-1* gene and all were found to express the mAb 3/1 epitope (Supplementary Data 1). In contrast, 16 of 19 environmental isolates from the same healthcare facility 12 to 21 years later (1997 to 2006) contained multiple independent mutations in *lag-1*

predicted to disrupt functionality. Specifically, a nonsense mutation (L48*), an insertion of a transposase, and the acquisition of a deleterious substitution (H28L) (Fig. 4b), each correlated with a lack of reactivity with mAb 3/1 from the Dresden panel classification (Supplementary Data 1). Consistent with our previous findings (Fig. 4a), the presence of a functional LPS *O*-acetyltransferase in this epidemiological cluster also correlated with resistance to killing in human plasma, whereas isolates with a non-functional *lag-1* were susceptible to killing, independent of the type of deactivating mutation (Fig. 4c). Of note, no clinical episodes of disease were identified to be caused by this epidemiological cluster after 1985, and all clinical isolates contained a functional *lag-1* gene. The identification of multiple independent mutations associated with loss of *lag-1* function suggests a selection pressure that drives the inactivation of the *lag-1* gene and LPS *O*-acetylation in this environment. The trend of *lag-1* gene loss or deactivation was not observed among longitudinal ST1 healthcare facility-associated isolates from a cluster sequenced in a recent study by David and colleagues[24]. However, a study on starvation of *L. pneumophila* in ultrapure water showed that in a short-term period, the viable cell numbers of all mAb 3/1-positive strains decreased strongly compared to the other strains suggesting a negative selection for *lag-1* function in some water environments[49]. Overall, these data indicate that the presence of a functional *lag-1* correlates with enhanced resistance to serum killing.

To investigate the role of *lag-1* in mediating resistance to serum killing, we introduced each of three major *lag-1* variants encoded on an expression plasmid (pMMB207) into the strain 4681 that contains a non-functional *lag-1* gene due to the insertion of a transposase. Introduction of each *lag-1* variant led to expression and LPS 8-*O*-acetylation as confirmed by flow cytometric detection of the mAb 3/1 epitope (Fig. 5a, b). Strikingly, complementation of the strains with each *lag-1* variant resulted in resistance to killing in human serum (Fig. 5d). In order to test this phenomenon in a distinct strain background, we introduced the *lag-1* variant 1 into the *lag-1*-negative strain 4312, which contains a 1 bp deletion resulting in a frameshift at position 156. Similarly, *lag-1* expression conferred resistance to killing in human serum (Supplementary Fig. 6). Consistent with this, we observed a decrease in human C3 deposition at the surface of the *lag-1* expressing bacteria when compared to the isogenic *lag-1*-negative strain, at levels similar to the wild-type *lag-1*-positive isolate 3656 (Supplementary Fig. 7). Of note, it was previously reported that a wild-type *lag-1*-positive isolate and a spontaneously derived *lag-1* mutant demonstrated similar levels of resistance to serum killing[20]. We were unable to obtain the relevant strains, but we speculate that the discordance with our findings could be due to phase-variable expression of LPS[19] or differences in transcriptional levels of *lag-1* gene that have been previously observed depending on culture conditions[50].

**lag-1 confers resistance to killing by the classical complement pathway.** We next investigated if a specific complement pathway was responsible for the serum killing of the Sg-1 mAb 3/1-negative *L. pneumophila*. As a similar inhibitory effect of *lag-1* on bacterial killing was observed in both serum and plasma, we employed serum for these experiments to facilitate the use of commercially available depleted serum samples. EDTA can be used to inhibit the classical, lectin and alternative pathways via chelation of both $Ca^{2+}$ and $Mg^{2+}$, whereas chelation with EGTA/$Mg^{2+}$ inhibits the $Ca^{2+}$-dependent classical and lectin pathways only, leaving the alternative pathway unaffected (Fig. 6a). Killing assays in human serum in the presence of either EDTA or EGTA/$Mg^{2+}$ resulted in complete abrogation of *lag-1*-negative strains

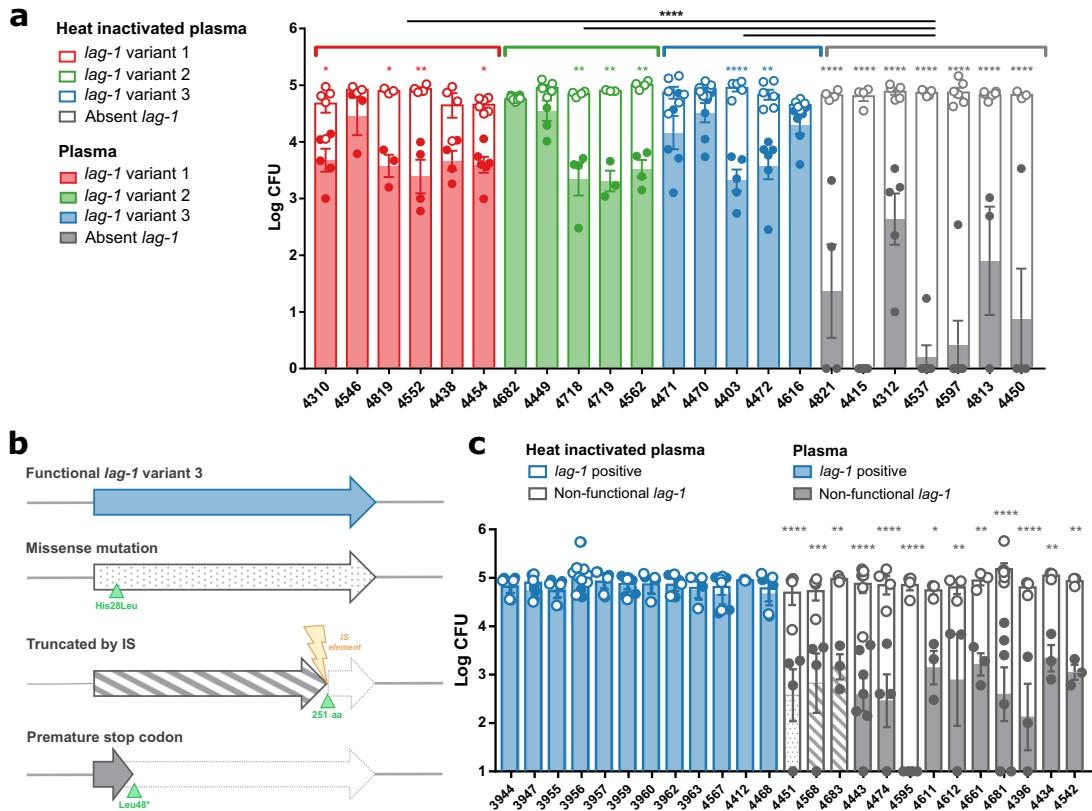

**Fig. 4 Presence of functional *lag-1* correlates with resistance to killing in human plasma. a** Representative *L. pneumophila* isolates containing allelic variants 1, 2, and 3 of *lag-1* (depicted in red, green, and blue, respectively) or *lag-1*-negative (depicted in grey) were incubated with human plasma (coloured bars and dots) or heat-inactivated plasma (open bars and dots) for 1 h at 37 °C. Each dot represents an average cfu count in plasma from a single donor (*n* = 7 donors for strain 4470, *n* = 6 for strains 4454, 4471, 4472, 4616, 4537, 4597; *n* = 5 for strains 4310, 4682, 4449, 4403, 4415, 4312; *n* = 4 for strains 4552, 4438, 4718, 4562, 4821, 4450 and *n* = 3 for strains 4546, 4819, 4719, 4813). Mean +/− SEM, One-way Anova with Holm-Sidak's multiple comparisons test between Plasma and Heat-inactivated plasma for the same strain. One-way Anova with Tukey's multiple comparisons test for differences between the variants. *\*p* < 0.05, *\*\*p* < 0.01, *\*\*\*\*p* < 0.0001. Source data and exact *p*-values are provided in the Source Data file. **b** Schematic representation of the multiple natural *lag-1* independent deactivating mutations in epidemiologically-related isolates. In blue—functional *lag-1*, with dots— *lag-1* with His28Leu mutation, with lines—*lag-1* truncated by an insertion sequence in aa 251, in grey—*lag-1* truncated by a premature stop codon. **c** Isolates with *lag-1* mutations show increased susceptibility to killing in human plasma. Closely related isolates from the same hospital-associated cluster were incubated with human plasma (coloured and patterned bars and dots) or heat-inactivated plasma (open bars and dots) for 1 h at 37 °C. Each dot represent an average cfu count in plasma from a single donor (*n* = 9 donors for strain 3956; *n* = 7 for strain 4443; *n* = 6 for strain 4681; *n* = 4 for strains 3947, 3959, 4567, 4451, 4474, 4568, 4595 and *n* = 3 for all other strains). Mean +/− SEM, One-way Anova with Holm-Sidak's multiple comparisons test between Plasma and Heat-inactivated plasma for the same strain, *\*p* < 0.05, *\*\*p* < 0.01, *\*\*\*\*p* < 0.0001. Source data and exact *p*-values are provided in the Source Data file.

susceptibility to complement, and no difference in cell viability compared to incubation in heat-inactivated serum, suggesting the alternative complement pathway is insufficient to kill *L. pneumophila* (Fig. 6b). To distinguish the role of the classical from the lectin pathway, killing assays were performed in the presence of mannose that competes for the association of mannose binding lectin (MBL) with the bacterial surface and blocks the lectin complement pathway[51]. Using this approach, there was no effect on bacteria viability, indicating that the lectin pathway is not required for *L. pneumophila* Sg-1 killing, consistent with the previous report that MBL polymorphisms are not associated with a higher risk for legionellosis[52]. The exclusion of the lectin pathway suggests the essential role of the classical pathway in the complement killing of *L. pneumophila*. To confirm this finding we depleted serum of C1q, required for the classical pathway, resulting in loss of serum-mediated killing (Fig. 6b). It is noteworthy that we also observed an increase in bacterial viability in the presence of Factor B-depleted serum (Fig. 6b), implying a possible role for the alternative pathway in amplification of classical pathway activation as previous described for

*Streptococcus pneumoniae*[53]. The impact of this pathway in *L. pneumophila* serum killing may be strain-dependent as indicated by the difference in the impact of Factor-B depletion on the 2 strains examined (Fig. 6b). Previously, purified *L. pneumophila* LPS was reported to activate both classical and alternate pathways, primarily through the activation of the classical pathway dependent on natural IgM antibodies[54]. In another study, complement C1q protein was demonstrated to bind to the major outer membrane protein (MOMP) of *L. pneumophila*, activating complement in an antibody-independent way[55].

**lag-1 expression enhances resistance to neutrophil phagocytosis, and survival in a murine model of pulmonary legionellosis.** Classical pathway complement activity exists in healthy human BAL, despite the relatively low concentration of some complement proteins[56], and exposure to aerosolized LPS leads to a rapid increase of the level of these proteins in the lung of human volunteers[57]. The importance of this innate immune mechanism in lung health is supported by the observation that many patients

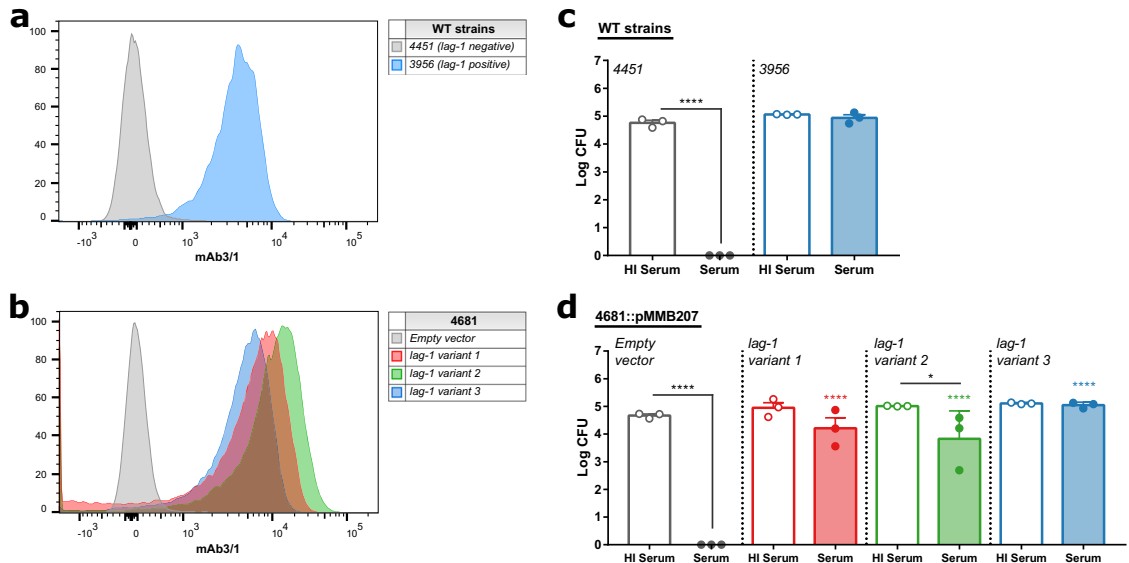

**Fig. 5 *lag-1* confers resistance to complement-mediated killing of *L. pneumophila*. a** and **b** Introduction of any *lag-1* gene variant leads to mAb 3/1 epitope expression. Detection of mAb 3/1 epitope by flow cytometry in (**a**) WT and (**b**) isogenic mutants of *L. pneumophila* isolates expressing the *lag-1* variant 1, 2 or 3. **c** and **d** *lag-1* expression confers serum complement resistance to *L. pneumophila* strains with non-functional *lag-1* gene. Isolates were incubated with human serum or heat-inactivated serum for 1 h at 37 °C. Each point represents an average of triplicate CFU counts of a single sera donor (*n* = 3). Bars represent mean + SEM. One-way ANOVA, Tukey's multiple comparisons test, *\*p* < 0.05, \*\*\*\**p* < 0.0001. Comparison to *Empty vector* isogenic strain represented on the top of the bars. Comparison to *Empty vector* isogenic strain represented on the top of the bars. Source data and exact *p*-values are provided in the Source Data file.

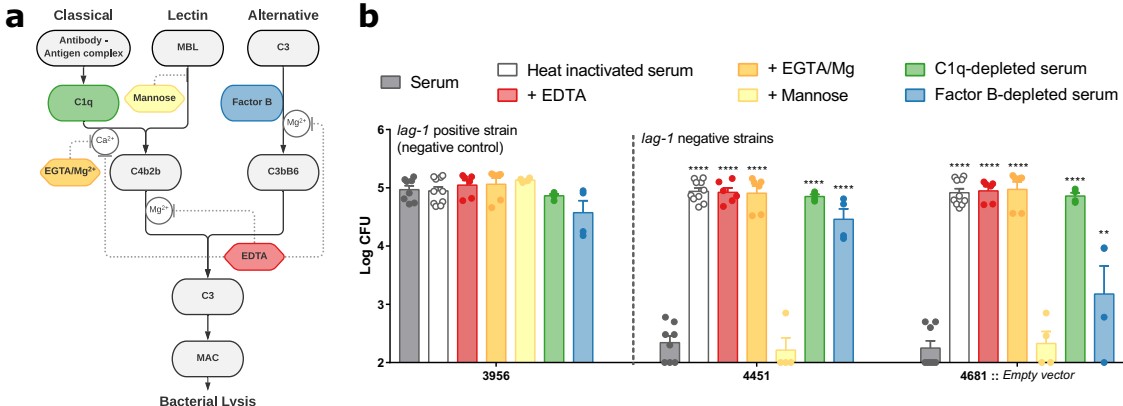

**Fig. 6 The classical pathway is essential for complement-mediated killing of *L. pneumophila*. a** Simplified schematic representation of the complement pathways and the inhibitors used in this study. Made using Lucidchart. **b** Inhibition of both classical and alternative pathways confers *L. pneumophila* resistance to complement killing in human serum. 3956, 4451, 4681::*Empty vector* isolates were incubated with 90% heat-inactivated serum, serum, serum with EDTA, EGTA/Mg²⁺ or mannose, Factor B- or C1q-depleted serum for a 1 h at 37 °C. Each point represents an average of three technical replicates for a single biological replicate (*n* = 8 biological replicates for heat-inactivated serum and serum; *n* = 6 for serum with EDTA and with EGTA/Mg; *n* = 4 for serum with Manose, Factor B- and C1q-depleted serum). Statistical analysis when compared to serum on top of the bars. One-way ANOVA, Tukey's multiple comparisons test, \*\**p* < 0.01 \*\*\*\**p* < 0.0001. Source data and exact *p*-values are provided in the Source Data file.

with deficiencies in complement proteins or complement receptors have recurrent respiratory infections[58]. The role of the complement system in lung immune defences extends beyond the proteolytic cascade associated with bacterial lysis, as opsonisation with complement C3b protein induces phagocytosis of opsonized targets by neutrophils and macrophages[59]. Accordingly, to test the effect of *lag-1* expression on phagocytosis of *L. pneumophila* by neutrophils, human blood purified neutrophils were incubated with fluorescein isothiocyanate (FITC)-labelled *L. pneumophila* strains in the presence of non-immune human serum (Fig. 7a to d). We examined naturally occurring *lag-1*-positive (*n* = 3) and *lag-1*-negative (*n* = 3) strains from diverse phylogenetic groups, and observed elevated FITC mean fluorescence intensity (MFI) in

neutrophils infected with *lag-1*-negative strains compared to *lag-1*-positive strains indicating a *lag-1*-dependent reduction in internalisation (Fig. 7a). Further, introduction of the *lag-1* gene on the pMMB207 expression plasmid significantly reduced neutrophil phagocytosis (Fig. 7c and Supplementary Fig. 6). The same effect was observed in neutrophils infected with the naturally occurring strains expressing DSred via pSW001 (Supplementary Figs. 8 and 9). Consistent with these findings, confocal microscopy of neutrophils infected with *lag-1* expressing strains had lower numbers of internalised bacteria when compared to neutrophils infected with *lag-1*-negative strains (Fig. 7b, d). Taken together, these data demonstrate a role for *lag-1* in resistance to neutrophil phagocytosis. It was previously shown that *lag-1*

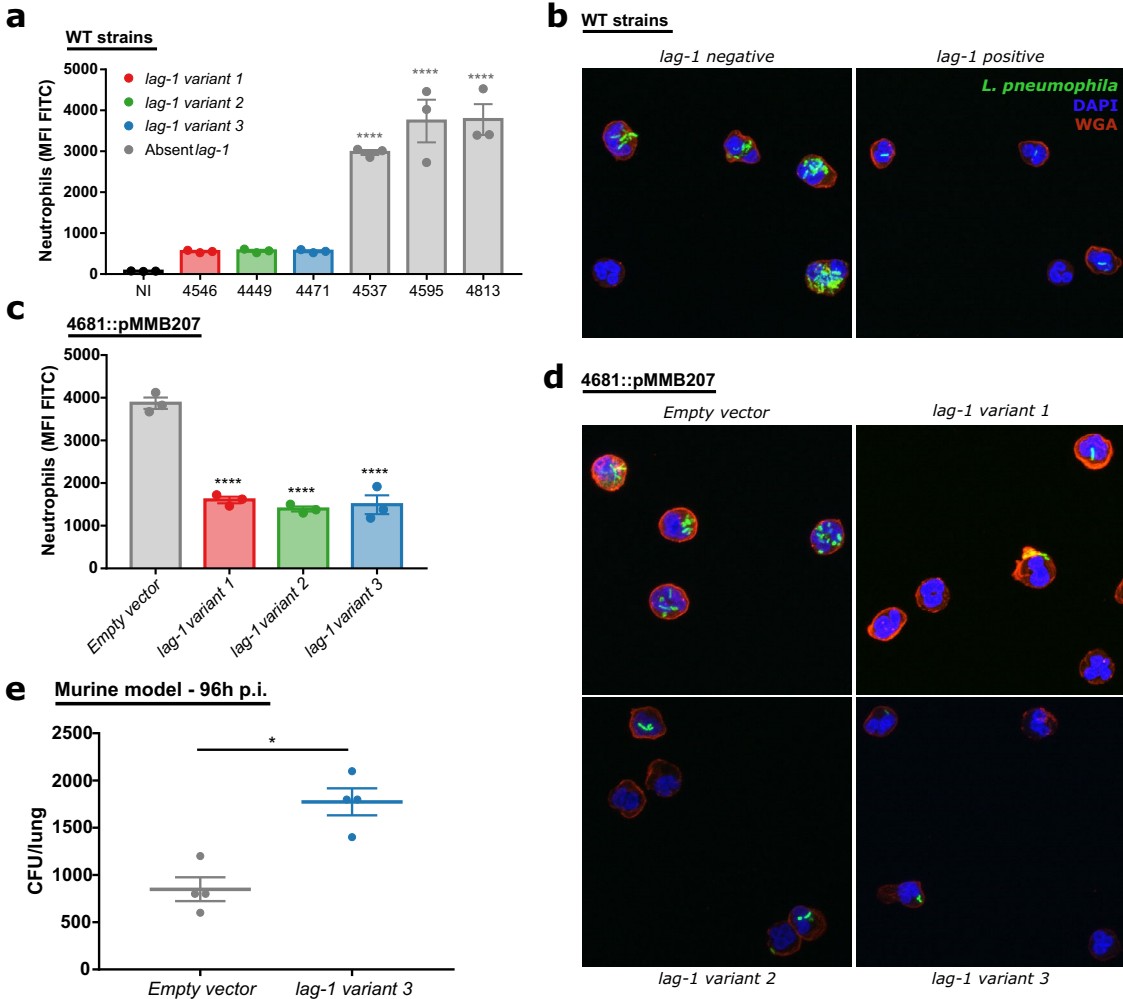

**Fig. 7 lag-1 confers resistance to human neutrophil phagocytosis and enhances surival in a murine model of pneumonia. a** to **d** lag-1 confers resistance to human neutrophil phagocytosis. WT lag-1-positive and -negative strains expressing DsRed fluorescent protein, as well as isogenic mutants of L. pneumophila isolates expressing the lag-1 variant 1, 2 or 3, were stained with FITC and pre-incubated with 10% human serum 15 min prior incubation with human neutrophils for 30 min. **a** and **b** Phagocytosis was evaluated by measuring neutrophils FITC fluorescence by flow cytometry. Each point represents a technical replicate for a single donor. Data are representative of three independent experiments. L. pneumophila strains expressing lag-1 variant 1, 2, or 3 are represented in red, green, or blue, respectively. Strains that do not express a functional lag-1 gene are represented in grey. Non-infected (NI) neutrophils are represented in black. Bars represent mean +/− SEM. One-way ANOVA, Dunnett's multiple comparisons test to non-infected control or to Empty vector, **$p < 0.01$, ****$p < 0.0001$. Source data and exact p-values are provided in the Source Data file. **c** and **d** Representative images of L. pneumophila-infected human neutrophils obtained by confocal microscopy and maximum intensity algorithm. Bacteria are represented in green, cytoplasmic membrane in red and nucleus in blue. **e** lag-1 expression increases L. pneumophila lung survival. C57BL/6J mice were intranasally infected with $1 \times 10^6$ L. pneumophila cells in 40 μL of PBS and bacterial numbers in the lungs evaluated 96 h post infection. Each point represents an individual animal ($n = 4$ in each group). Data are presented as mean +/− SEM. Two-tailed Mann–Whitney t-test *$p < 0.0286$. Source data are provided as a Source Data file.

expression is associated with an increase in ability of L. pneumophila to adhere and, consequently, penetrate into Acanthamoeba castelani membranes[60]. However, the ability of LPS to inhibit phagosome maturation in A. castellanii was reported to be independent of a functional lag-1[61]. Our data suggest that that internalisation by neutrophils is inhibited by lag-1 and a requirement for serum suggests an effect that is complement-dependent (Supplementary Fig. 9). Therefore, the ability of lag-1-positive L. pneumophila strains to escape complement deposition correlates with the capacity to escape phagocytosis. These data are consistent with the findings of early L. pneumophila studies, where the L. pneumophila strain Philadelphia 1 (lag-1-positive) was demonstrated to be resistant to complement and neutrophil killing in the absence of specific antibodies[62,63]. Of note, there is

increasing evidence for the pivotal role of neutrophils in the resolution of L. pneumophila lung infections, and either neutrophil depletion or blockage of recruitment to infected lungs renders mice susceptible to L. pneumophila[64−68].

In order to test the role of lag-1 in the pathogenesis of legionellosis in vivo we used an established murine model of pneumonia[69]. C56BL/6 mice were intranasally infected with L. pneumophila strain 4681 containing either the empty plasmid or plasmid expressing the lag-1 variant 3. Mice infected with the lag-1-expressing strain exhibited higher bacterial burden in the lungs at 96 h post infection compared to mice infected with the lag-1-negative isogenic strain (Fig. 7e). These data indicate that acquisition and expression of lag-1 enhances L. pneumophila survival during infection of the lung.

Taken together, our data indicate that *L. pneumophila lag-1* mediated inhibition of complement deposition, disrupts the innate immune response via inhibition of complement-mediated lysis and inhibition of neutrophil phagocytosis, promoting survival in vivo.

## Conclusion

The steady global increase in *L. pneumophila* infections is worrisome and studies that explore the evolutionary basis of increased pathogenicity can provide insights into the nature of the public health threat posed. Here we have employed a combined large-scale population level study of clinical and environmental isolates, along with functional ex vivo and in vivo analyses to reveal the evolutionary and functional basis for serum resistance in *L. pneumophila*. The apparent selective advantage conferred by *lag-1* for disease in humans is intriguing in the context of its environmental reservoir. It is feasible that *lag-1* provides a functional advantage in its natural amoebal host as reported by Palusinska-Szysz et al., and that co-selection occurs for enhanced pathogenicity in humans[60]. Alternatively, David et al previously proposed that the recent expansion of *L. pneumophila* in man-made water systems and global spread of selected clones is consistent with possible human-to-human or human-to-environmental dissemination[22]. If indeed some *L. pneumophila* clones are evolving towards human colonisation and transmission, *lag-1* may represent a key human host-adaptive trait. Finally, we suggest that the identification of a specific modification of LPS that is required for resistance to classical complement-mediated killing, inhibits phagocytosis by neutrophils and promotes survival in vivo, could inform the design of novel therapeutic approaches that subvert the capacity of *L. pneumophila* to resist innate immune killing during severe human infection.

## Methods

**Bacterial culture**. *L. pneumophila* Sg-1 isolates from the *Scottish Haemophilus, Legionella, Meningococcus & Pneumococcus* Reference Laboratory (SHLMPRL), Scottish Microbiology Reference Laboratories (SMiRL), Glasgow collection were grown on buffered charcoal yeast extract plates (Legionella CYE agar base-Oxoid, BCYE Growth Supplement SR0110-Oxoid) for 3 days at 37 °C. Three colonies were inoculated is of supplemented yeast extract broth (YEB, Yeast extract—Fluka, BCYE Growth Supplement SR0110-Oxoid) for 24 h at 37 °C 200 rpm. Strains were subcultured by transferring 150 µL of the previous inoculation into 15 mL of YEB at 37 °C 200 rpm until it reaches O.D. 600 nm ≈ 0.8. When necessary, media was supplemented with 10 µg/mL of chloramphenicol (Sigma-Aldrich) for plasmid maintenance. After submission of the first version of the manuscript, the BCYE Growth Supplement SR0110 from Oxoid was discontinued. The data from Supplementary Fig. 10 was obtained during revision with bacteria cultured with the alternative BCYE Growth Supplement LS0053 (E&O Laboratories Limited) to post-exponential phase O.D. 600 nm ≈ 1.4, and on *Legionella* BCYE with GVPC pre-poured plates (PP0870, E&O Laboratories Limited). *Escherichia coli* cells were cultured at 37 °C in Luria-Bertani (LB) broth or agar (Melford) supplemented with 10 µg/mL of chloramphenicol (Sigma-Aldrich) or 100 µg/mL of ampicillin (Sigma-Aldrich). For blue-white colony screening, LB agar plates were spread with 2% X-gal (Melford).

**Whole-genome sequencing**. Four-hundred seventy-four *Legionella* isolates from the SHLMPRL isolate collection were recovered from frozen stocks and sent to MicrobesNG (microbesng.uk, Birmingham, United Kingdom) for DNA extraction and whole-genome sequencing as follows: plated cultures of each isolate were inoculated into a suspension of plastic beads in a cryopreservative (Microbank™, Pro-Lab Diagnostics UK, United Kingdom). Three beads were washed with extraction buffer containing lysozyme (final concentration 0.1 mg/mL) and RNase A (ITW Reagents, Barcelona, Spain) (final concentration 0.1 mg/mL), incubated for 25 min at 37 °C. Proteinase K (VWR Chemicals, Ohio, USA) (final concentration 0.1 mg/mL) and SDS (Sigma-Aldrich, Missouri, USA) (final concentration 0.5% v/v) were added and incubated for 5 min at 65 °C. Genomic DNA was purified using an equal volume of SPRI beads and resuspended in EB buffer (Qiagen, Germany).

DNA was quantified with the Quant-iT dsDNA HS kit (ThermoFisher Scientific) assay in an Eppendorf AF2200 plate reader (Eppendorf UK Ltd, United Kingdom). Genomic DNA libraries were prepared using the Nextera XT Library Prep Kit (Illumina, San Diego, USA) following the manufacturer's protocol with

the following modifications: 2 ng of DNA were used as input, and PCR elongation time was increased to 1 min from 30 s. DNA quantification and library preparation were carried out on a Hamilton Microlab STAR automated liquid handling system (Hamilton Bonaduz AG, Switzerland). Pooled libraries were quantified using the Kapa Biosystems Library Quantification Kit for Illumina on a Roche light cycler 96 qPCR machine. Libraries were sequenced with the Illumina HiSeq using a 250 bp paired end protocol.

**Sequence processing and analysis**. Reads were trimmed with Trimmomatic (v0.36) using default settings and de novo assembly was performed using SPAdes (v3.10.0) and contigs were reordered with Mauve Contig Reorderer (Muave v2.4.0)[70]. Gene annotation and functional prediction for each assembly was generated using Prokka (v1.12). The quality of whole-genome sequencing was assessed using Quast (v4.5) and genomes that exceeded the following metrics were selected as high-quality assemblies for analyses. (i) Minimum 1.5 Mbp aligned to the *L. pneumophila* Philadelphia reference (total aligned length), (ii) Maximum duplication ratio of 1.03 (based on distribution), (iii) Minimum total length of contigs larger than 50 Kbp is 1 Mbp. To maximise the fraction of core genome that can be aligned, isolates belonging to the minor subspecies, (*L. pneumophila* subsp. *fraseri*, *L. pneumophila* subspecies *pascullei* and *L. pneumophila* subspecies *raphaeli*) were not included in downstream analyses. Whole-genome sequences are available on the European Nucleotide Archive (ENA) database under Bioproject number PRJEB31628 [https://www.ebi.ac.uk/ena/browser/view/PRJEB31628]. Functional prediction of non-synonymous mutations in *lag-1* were predicted using PROVEAN (v1.1.3)[71].

Twelve genes from the LPS cluster, specific to Sg-1 and overlapped by *k*-mers enriched in clinical isolates (*lpg0762*, *lpg0766*, *lpg0767*, *lpg0768*, *lpg0771*, *lpg0772*, *lpg0773*, *lpg0777*, *lpg0778*, *lpg0779*, *lpg0780*, *lpg0781*). Orthologs of these 12 genes were identified. Genes that were not conserved in all 23 genomes analysed were removed (*lpg0771* and *lpg0777*). Paralogs of *lpg0778* were also removed. Genes were translated and aligned at the codon level using TranslatorX and Mafft. Alignments for all 12 genes were concatenated. Phylogenetic networks were generated using uncorrected P-distances with the equal angle method in Splitstree v4.14.6[72].

Sequence alignments were visualised in Artemis Comparison Tool v.17.0.1 and figures generated with Easyfig v2.2.2[73,74].

**Genome-wide association analyses**. Five-hundred five high-quality publicly available genome assemblies of *L. pneumophila* subsp. *pneumophila* were downloaded on the 2nd of March 2017 for inclusion in these analyses. Quality criteria for inclusion of published genomes: fewer than 200 contigs of 1 kb or longer, less than 1.03 duplication ratio, not a subspecies or taxonomic outlier (bright green or grey clade), contigs >50 Kb add up to at least 1 Mbp, not >700 uncalled bases per 100 Kb. At least 3.2 Mbp in length.

From the combined dataset of 902 isolates, we were able to extract metadata describing the isolation source for 758 isolates (environmental or clinical). A ML phylogenetic tree was then constructed using RAxML (v8.2.10) from the 1.8 Mbp core genome alignment generated using Parsnp v1.2 (https://github.com/marbl/parsnp). This phylogenetic tree was used as input for the subsampling approach. An automated subsampling algorithm was used to reduce the redundancy of the dataset by iteratively removing one of each pair of isolates, which had the same source type. We implemented this algorithm in a modified version of a phylogeny-based dataset reduction tool called Treemmer (downloaded: 8th March 2018)[75]. The modified subsampling script is available on github.com/bawee/Treemmer. Specifically, we added the ability to take into account phenotypic categories (i.e., clinical or environmental) before iteratively removing one taxon from each pair of the most closely related taxa in the phylogeny. This was repeated until the minimum distance between the pairs of isolates with the same phenotype reached a user-defined threshold. This approach was performed with increasing minimum patristic distance thresholds between pairs on the tree, ranging from 0.0001 (≈180 SNPs), 0.001 (≈1800) and 0.01 (≈18,000 SNPs). The subsampled datasets contained 452 (60%), 382 (50%) and 353 (47%) of the original isolates, respectively.

GWAS analysis based on *k*-mer enrichment was performed using SEER (v1.1.3alpha)[76]. The frequency of *k*-mers between 12 and 100 bp long were counted with fsm-lite v1.0 (-m 12 -M 100). Only *k*-mers present in 5% and 95% of the total number of isolates (-s < 5%> -S < 95%>) were used and -maf 0.05 (minimum allele frequency) was used when performing the SEER analysis. The frequency and distribution in clinical and environmental isolates were evaluated individually for each *k*-mer. Pan-GWAS was performed using the homologue clusters generated by ROARY (v3.12.0) using the following settings (-i 95 -s)[32]. The association analyses were calculated using SCOARY (v1.6.9)[33]. The following thresholds were used to identify the highest-scoring associations: Naive *p*-value < 0.05, *p*-value after Bonferroni correction <0.05, Benjamini–Hochberg *p*-value < 0.05 and Empirical *p*-value after 500 label-switching permutations <0.05 (-e 500).

**Molecular biology**. For *lag-1* cloning, primers (Variant 1: Fwd- GGC **CGA ATT** CGT AAG GAA AAA TAA TTT ATC, Rev- CCC **GGA TCC** TTA TGT TGA ATA AGC TAA CTT GTT TGA TGT; Variant 2: Fwd – GGC **CGA ATT** CGT AAG GAA AAA TAA TTT ATC, Rev – AT**G GAT CC**T TAC ATC ATC ACC ATC ATC ATT GTT GAA TAA GCC AAC; Variant 3: Fwd – GGC **CGA ATT** CAC ATG CAA GAA TAA TTTA, Rev – CCC **GGA TCC** GCT TAC GTA ATA

TAA GCT AAC TTA TTT GAT GTG*Eco*RI and *Bam*HI restriction sites in bold) were used to amplify a 1221 bp fragment of chromosomal DNA from *L. pneumophila* strains 4454, 4449 and 4471, encoding the variant 1, 2 and 3 of *lag-1* allele and upstream region, respectively. These fragments were blunt cloned with StrataClone Blunt PCR Cloning kit (Agilent) into vector pSC-B and transformed into *E. coli*. Plasmids from transformants were purified with Monarch Plasmid DNA Miniprep Kit and digested with *Eco*RI and *Bam*HI enzymes (New England BioLabs). *lag-1* gene was ligated to pMMB207 using T4 DNA ligase (New England BioLabs) and both pMMB207::*lag-1* and empty plasmid were electroporated into the electrocompetent mAb 3/1-negative strains 4681 and 4312 using a Gene Pulser II (Bio-Rad) following published protocols[77]. To confirm *O*-acetyltransferase activity of the *lag-1* cloned bacteria, transformed and positive control strains were fixed in 10% formalin (VWR) for 20 min and incubated with 1:100 of mouse mAb 3/1 (Dresden panel) for 30 min in ice prior incubation with 1:100 polyclonal AlexaFluor 488 conjugated donkey anti-mouse IgG (H + L) (Cat no. A21202, ThermoFisher Scientific). After incubation on ice for 30 min, the bacteria were washed and resuspended in PBS for flow cytometry analysis. DsRed fluorescent protein expression was introduced into WT *lag-1*-positive and -negative strains by electroporation with pSW001 plasmid (pMMB207C, ΔlacIq, constitutive *dsred*)[78]. DsRed expression was confirmed by flow cytometry. Formaldehyde-fixed samples were analysed on a BD LSRFortessa X-20 flow cytometer (BD biosciences) and data analysed using FlowJo software (v10).

**Ethics statement.** Ethical approval for the collection of blood from anonymous donors was granted by the University of Edinburgh Research Ethics Committee. This study was reviewed by the University of Edinburgh, College of Medicine Ethics Committee (2009/01) and subsequently renewed by the Lothian Research Ethics Committee (11/AL/0168). Written informed consent was received from all volunteers participating in the study. No compensation was provided to the volunteers. All animal experiments were approved by governmental animal welfare committees by the LaGeSo (Landesamt für Gesundheit und Soziales) Berlin. All experiments are assigned to project G0334/17.

**Plasma and serum killing assays.** Human blood was obtained from healthy volunteers in syringes treated with anticoagulant citrate dextrose. Plasma was obtained by centrifuging whole blood for 15 min at 25 °C at 1200 × g without the centrifuge brake and collecting the layer above the buffy coat. Normal serum was collected from human blood obtained from healthy volunteers in BD Vacutainer serum tubes. Tubes were centrifuged for 10 min at 25 °C 1200 × g and the supernatant collected. Plasma and serum were stored following immediate freezing at −80 °C. Samples used were negative for anti-*L. pneumophila* antibodies as determined by standard diagnostic serology methods (immune fluorescence test). The ability of *L. pneumophila* strains to resist killing by human plasma and serum was determined by incubating $1 \times 10^5$ bacterial cells in 90% plasma or serum at 37 °C for 60 min. Serial dilutions were plated in duplicate on BCYE media, incubated at 37 °C for 3 days before enumeration. Heat inactivation of plasma was carried out by incubation at 56 °C for 30 min in a water bath. Both plasma, serum, heat-inactivated plasma and heat-inactivated serum were centrifuged at 4000 × g for 20 min at 4 °C prior use. Inhibition of complement pathways was carried out by adding 12.5 mM EDTA (Sigma-Aldrich), 12.5 mM EGTA/Mg$^{2+}$ (Sigma-Aldrich) or 100 mM mannose (Acros Organics) to normal human serum or by using C1q- and Factor B-depleted serum (Pathway Diagnostics). GraphPad prism v7 was used for statistical analysis.

**C3 binding assays.** For analysis of C3 deposition on the surface of *L. pneumophila*, bacteria were grown to post-exponential phase and fixed in 10% formalin (VWR) for 20 min. Nunc maxisorp ELISA plates were coated with $1 \times 10^7$ bacteria per well overnight at 4 °C, blocked with 1% w/v bovine serum albumin (Sigma-Aldrich) in PBS for 2 h and incubated with serial dilution of human serum for 2 h at room temperature. Binding of C3 at the surface of the bacteria was determined by incubating with 1:100 FITC conjugated goat F(ab')2 anti-human C3 complement (Protos Immunoresearch, Cat no. 365) for 2 h at room temperature and fluorescence detected in using the CLARIOstar fluorescence plate reader (BMG Labtech). GraphPad prism v7 was used for statistical analysis.

**Neutrophil phagocytosis assay.** Human venous blood was obtained from healthy volunteers in syringes treated with anticoagulant citrate dextrose and the isolation of neutrophils was performed by Ficoll/Histopaque centrifugation[79]. *L. pneumophila*-DsRed strains and *L. pneumophila* isogenic strains were labelled with 0.5 mg/mL of FITC (Sigma) solution for 30 min on ice in the dark. After several washes with PBS to remove any non-associated FITC, the bacteria were incubated with 10% normal human serum for 15 min at 37 °C before incubation with $2.5 \times 10^5$ purified neutrophils (MOI 10). Infection was maintained at 37 °C, 5% CO$_2$ with orbital shaking at 600 rpm for 30 min. Cells were then fixed in 10% formalin (VWR) for 20 min and kept in PBS for flow cytometry analysis. Samples were analysed on a BD LSRFortessa X-20 flow cytometer using the FACSDiva software v9 (BD biosciences) and data analysed using FlowJo software (v10). Neutrophils were selected by their FSC/SSC profile and doublets excluded using FSC-H/FSC-A ratios. MFI was calculated in the total single cell population (Supplementary Fig. 9). GraphPad prism v7 was used for statistical analysis.

**Confocal microscopy.** *L. pneumophila*-infected neutrophils were incubated for 20 min with WGA-AF555 (Invitrogen) and with Hoechst 33342 (Thermo Scientific) for 5 min for cytoplasmatic membrane and nucleus staining, respectively. Cells were attached to a microscope slide using cytospins and analysed using the Zeiss LSM 710 inverted confocal. The images were obtained using a maximum intensity projection algorithm.

**Murine pulmonary infection.** All mice used were 12–14 weeks old, female WT on a C57BL/6J background. Mice were either purchased from Charles River or from institutional breeding stock at the Charité-University Medicine, Berlin. Upon transfer to the animal unit, mice were kept in ventilated cages at a conditioned room temperature at 22 °C +/− 2 °C and humidity at 55% +/− 5%. A dark/light cycle of 12 h/12 h was maintained. All experiments were approved by the LaGeSo (landesamt für Gesundheit und Soziales) Berlin. Anaesthetised mice were intranasally infected with $1 \times 10^6$ *L. pneumophila* in 40 μL of PBS. 96 h post infection, mice were anesthetised and euthanized through final blood withdrawal. Lungs were flushed via the pulmonary artery with sterile 0.9% NaCl, subsequently removed and homogenised using a cell strainer (100 μM, BD Bioscience). For determination of bacterial counts, the lung homogenates were lysed with 0.2% Triton X-100. Serial dilutions were plated on BCYE agar and incubated for 3 days at 37 °C.

**Reporting summary.** Further information on research design is available in the Nature Research Reporting Summary linked to this article.

## Data availability
The sequence data that support the findings of the study have been deposited in the European Nucleotide Archive (ENA) database under Bioproject number PRJEB31628 [https://www.ebi.ac.uk/ena/browser/view/PRJEB31628]. Accession numbers of each genome used is listed in Supplementary Data S1. A source data file is provided with this paper. Source data are provided with this paper.

## Code availability
We have implemented a modified version of the subsampling software Treemmer (https://github.com/fmenardo/Treemmer)[75], which is available at http://github.com/bawee/Treemmer.

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

## Acknowledgements

This work was supported by funding to JRF from the Chief Scientists Office Scotland (Grant No ETM/421), the Wellcome Trust Collaborative award (Grant No. 201531/Z/16/Z), the Medical Research Council (UK) award MR/N02995X/1, and Biotechnology and Biological Sciences Research Council institute strategic grant funding (ISP2) (Grant no. BB/P013740/1). Computing resources were supported in part by MRC CLIMB (Grant Number: MR/L015080/1). A.B.K. was supported by the International Max-Planck Research School (IMPRS-IDI), and B.O. by the German Research Foundation (OP 86/12-1 and SFB-TR84A1/A5). We are grateful to Prof. Carmen Buchrieser for providing plasmids pMMB207 and pSW001. Thanks also to Prof. Kenneth Baillie, Dr. Sara Clohisey-Hendry and Dr. Clark Russell for organising the human blood donation study and the volunteers from the Roslin Institute who provided blood samples for the serum and plasma killing assays and neutrophil assays.

## Author contributions

B.A.W. designed and performed the population genomic analysis and wrote the paper and J.A. designed and performed the functional experiments, analysed data and wrote the paper; D.S.J.L. provided the strains, carried out bacterial subtyping and conceptualised the study, A.B.K. performed the mouse experiments, F.A.S performed functional experiments, A.P. provided scientific and technical support for cloning, J.G. analysed data, R.L.C., J.C. and P. M. developed analytical tools, B.O. designed the murine infection studies, A.J.S. provided strains and conceptualised the study, J.R.F. conceptualised the study, analysed data and wrote the paper.

## Competing interests

The authors declare no competing interests.
