## [Peer Review File · Nature Communications]

Population analysis of *Legionella pneumophila* reveals a basis for resistance to complement-mediated killingREVIEWER COMMENTS

Reviewer #1 (Remarks to the Author):

"Population analysis of *Legionella pneumophila* reveals the basis for resistance to complement-mediated killing" by Wee and Alves et al.

In this work, the authors aim to answer why some *L. pneumophila* are associated with clinical disease in humans while others are not, a feature which to only some degree is dictated by clonal association. They do this by gathering a diverse collection of clinical and environmental samples and perform a genome-wide association study to investigate elements that are enriched in the clinical samples, and additionally validate their findings with functional assays.

I was quite impressed with the "soup to nuts"-approach methodology here - The authors go from collecting samples, whole-genome sequencing, in silico analysis, to wet lab transformation, in vitro functional validation of their findings and even in vivo in mice. The standard of evidence is really high here and it is obvious that the paper actually delivers on the promise from the title, i.e. the basis, at least partly, for complement resistance. I wish more functional studies followed this example! I am not competent to evaluate the wet lab work described here, but the bioinformatic work is performed according to a gold standard and the figures are useful and to the point. In short, it's a solid paper without any significant technical or logical shortcomings.

I would only like to raise two minor issues related to reproducibility:

- I don't think the results could be reliably reproduced since Treemmer has a random component, and as far as I could tell there is no indication of which isolates were selected in each pruning step.
- Most of the cited software are lacking version numbers.

Reviewer #2 (Remarks to the Author):

The article « Population analysis of *Legionella pneumophila* reveals the basis for resistance to complement-mediated killing » by Wee and colleagues report a population genomics analyses based on 902 *Legionella pneumophila* strains with the aim to identify factors important for human infection. Genome sequencing of 397 Scottish *L. pneumophila* strains isolated from clinical cases or the environment was undertaken for this study, the remaining strains had been retrieved from public databases. Based on a GWAS analyses the authors report that the cluster coding for the LPS biosynthesis of SG-1 was highly enriched, and one gene of this cluster, *lag-1* showed the strongest association with clinical isolates. The authors further showed that recombination led to the dissemination of different alleles of *lag-1* but that three are dominant in the population. Based on an early study of nosocomial *L. pneumophila* that reported that this strain survived complement killing compared to an environmental strain, the authors set out to analyze this phenotype and to investigate whether it is conferred by the presence of *lag-1* in the genome. Using human blood from different donors, they tested representatives of the three different *lag-1* carrying strains and strains lacking *lag-1* with respect to serum killing. The authors show that *lag-1* coding strains had higher resistance to killing in human plasma and that naturally occurring *lag-1* inactivating mutants were less resistant. Using different inhibitors of the complement pathway they then further showed that *lag-1* confers resistance to complement mediated killing. Finally, using human neutrophils and a mouse model of pulmonary infection, the authors report that *lag-1* confers resistance to phagocytosis by human neutrophils and enhances virulence in pulmonary mouse infection.

The authors present a large body of work that seems to be well analyzed and interpreted and that is nicely illustrated with very comprehensive figures. The article is well presented and provides new data for a better understanding of the higher association of specific clones of *L. pneumophila*

with human disease, however, some points need further clarification.

General comments:

The authors divide the strains they analyze into clinical and environmental strains and find a high association of the lag-1 gene with clinical strains. However, although clinical strains can be clearly defined as they caused disease, this is not so clear for "environmental" strains in particular when relying on strain information obtained from Public database deposited strains. Furthermore, several of the environmental strains have been collected when searching for outbreak sources and are labelled environmental. How the authors can ascertain that the strain definitions "clinical" and "environmental" are well assigned, as this definition is the basis for their GWAS analyses?

I am confused with respect to the evolutionary history of lag-1. *L. pneumophila* are accidental, human pathogens, they are not transmitted from humans to humans, evolution for being "more virulent" does not take place in humans as the selection pressure is the environment, protozoa. Thus, how an evolution for enhanced resistance to serum killing should take place? There seems even be a negative selection for lag-1 in the environment as stated also by the authors (lines 268-269). How the authors explain this selection of specific lag-1 alleles for being associated to human infection?

Why many strains carry non-functional pseudogenes of lag-1? (lines 199-200) but at the same time the authors state that there is a strong selection pressure for lag-1 function? (lines 201-203). Could the authors speculate what the role of lag-1 is in the environment?

I am puzzled by the describing of the assays defining serum resistance of the different strains. According to the M&M section bacteria were grown on plate, then precultured, new medium was inoculated and then a culture harvested at OD₆₀₀ = 0.8 was used for assessing resistance to killing by human plasma or phagocytosis by neutrophils. *L. pneumophila* is known to have a biphasic life cycle and only bacteria grown to post exponential phase (e.g. OD 3.5 depending on the conditions) are known to be virulent and infectious. Thus bacteria at OD=0.8 are far from being in the PE phase but are in exponential phase. Could the authors explain? The authors should test PE phase bacteria if they give the same result.

Minor comments:

Line 47, reference 1 is a reference from 2011, there are several newer ones that should be added or cited in place. e.g. Boamah et al., *Front cell Infect Micorbiol* 2017; Mondino et al. *Annu Rev Pathol.* 2020

Line 57, please add reference 11 here

Line 119 please add a reference for these data

Line 132 it reads... "The factors that contribute to the enhanced human pathogenic capacity of some *L. pneumophila* clones are unknown"..... it could also be enhanced infectivity of transmissibility

Lines 148 – 153 the genes that were determined by GWAS analyses to be highly associated to human disease clones were identified as the LPS gene cluster of Sg1... please add the reference Cazalet et al. *Genome Research* 2010, and discuss that the LPS cluster has also been identified in this comparative genome analyses using microarrays, as associated to human disease.

Lines 182 -... where the different alleles of lag-1 are described I was missing an alignment of the three major alleles to appreciate the differences. Please add a definition on what basis the different alleles were defined.

Supplementary Table 3a: please add a Table legend to explain what the different parts of the Table show.

Line 211, please add reference David et al., PLoS Genet. 2017 Jun 26;13(6):e1006855. Here recombination among LPS gene clusters has already been described and Cazalet et al 2010, who described HGT of the LPS cluster among strains.

Line 227-230 it reads" A comparison of re-assortment rates across major phylogenetic lineages showed no significant differences between LPS and non-LPS genes across the *L. pneumophila* genome ($p=0.6275$), suggesting that recombination is active on a genome-wide scale " This has already been shown previously in David et al., PLoS Genet. 2017 Jun 26;13(6):e1006855 and Gomez-Valero et al., BMC Genomics 2011 Nov 1;12:536. please mention and cite these references.

Reviewer #3 (Remarks to the Author):

The study by Wee et al. investigates the genetic basis of pathogenicity in the bacterial pathogen, *Legionella pneumophila*, using combined genomic and experimental approaches. This is an important area of research since our lack of understanding to date of why some *L. pneumophila* strains are over-represented among disease cases has constrained our ability to improve various aspects of disease control. New knowledge has the potential to enable more targeted environmental risk assessment and therapeutic strategies, and will also shed light on how this environmental bacterium has evolved to become a cause of serious disease.

Using a genome-wide association study (GWAS) with a large collection of clinical and environmental isolates, the authors identify a single gene, *lag-1*, involved in LPS modification, to be strongly associated with clinical isolates. They show that *lag-1* has been distributed horizontally across all major clades of the species by recent recombination events. Using experimental approaches, they demonstrate the mechanism by which *lag-1* increases pathogenicity - by inhibiting complement-mediated killing and reducing phagocytosis by human neutrophils. Finally, they show that *lag-1* also promotes survival in a murine model of pulmonary legionellosis. Overall this is a beautiful story with an exciting finding, and provides an important advance in our understanding of *L. pneumophila* evolution and pathogenicity. The genomic analyses (for which I have been specifically asked to comment) have been performed to the highest standard, and generally the manuscript is clearly presented and contains detailed methodology. As such, I only have minor comments and suggestions listed below, as well as a more general discussion point.

Lines 97-98. Are the clinical isolates from Legionnaires' disease patients (presumably), or are any from Pontiac fever patients? This seems relevant given the focus on pathogenicity, but may not be obvious to a non-*Legionella* audience.

Lines 98-99. If some of the environmental isolates are outbreak-related, it might be worth a comment on how this would affect the GWAS?

Lines 124-126. This assumes that the *L. pneumophila* obtained from cooling towers and other artificial environments are representative of environmental *L. pneumophila*. The wording could be tweaked to something like...

"Notably, 35% (81 of 231) of environmental isolates belong to one of the five clinically dominant STs suggesting that at least a third of all environmental *L. pneumophila* *from the sampled sources* in Scotland have human pathogenic potential."

Line 145. Would it be useful to mention here that the Philadelphia genome is *sg-1*?

Lines 148-152. Are these genes with significant k-mer hits, including *lag-1*, specific to *sg-1*?

Lines 160-161.

"...this approach also indicated that the LPS biosynthesis genes were enriched among clinical isolates..."

Would it be clearer to say “sg-1 LPS biosynthesis genes”? It becomes clear that you mean that the sg-1 LPS genes are enriched (rather than LPS genes more generally) in both the SEER and Scoary analyses, but I think this could be clarified more here and in the preceding text (this relates to the previous two comments).

Lines 131-180. Do you know why other LPS genes are coming up in the GWAS? Is it because these are sg-1 genes that are more likely to be found with lag-1, or could these be important independent of lag-1?

Lines 131-180. Having identified lag-1 through statistical methods, it could be useful to then provide the number of clinical vs environmental genomes that lag-1 is found in. It might also be striking to see the clinical/environmental and lag-1 presence/absence rings on the same tree, but appreciate too that this might look more messy.

Lines 200-203. It’s fascinating that the lag-1 gene appears to have spread recently over different clades of *L. pneumophila*. I wonder whether the evidence for recent horizontal spread could be improved by putting the amount of lag-1 variation into context, i.e. how much lower is the allelic variation in lag-1 than in other core genes? Also, how long is the lag-1 gene? (I would be more convinced that identical sequences indicate recent horizontal spread if the gene is not very small.) It would also be interesting to know how different the three major variants are from each other.

Lines 446. “not more less than 200 contigs of more than 1Kb.”
I think this needs correction?

Lines 457. github.com/bawee/Treemmer doesn’t seem to be available.

Figures 1 and 3. The tree scale should be described in the legend.

Table S3. Do the values in the table refer to % nucleotide divergence?

Is the discovery of a gene that enhances the ability of *L. pneumophila* to infect humans (and mice) and which has spread rapidly over recent years not a very intriguing finding, also in light of the recent expansion of major disease-associated clades? Why would this be the case? Previous work has shown that multiple different *Lp* strains have spread rapidly around the world and found themselves in lots of different man-made environments (as evidenced from the diversity of places that people become infected). The repeated, recent acquisition of lag-1 that you show has occurred in many of these different lineages likely suggests one common selection pressure. Perhaps that selection pressure is due to one common amoebal species that is present in all these different man-made environments, but couldn’t it also be possible that the one shared feature is human exposure and these strains are adapting to (mostly asymptomatic/mild) human infection? That also avoids the need for additional explanation of how an adaptive trait relating to amoebae survival/infection (or other hosts/conditions) also co-incidentally results in increased virulence in humans. I appreciate that the authors say that the selective advantage of lag-1 is “unclear” and thus leave any other speculation on this matter to their own discretion!

What’s also striking to me is that the phylogenetic distribution of lag-1 suggests that its acquisition may have driven some of the very recent clonal expansions (e.g. most clear for ST47 and ST42), likely in combination with other factors. Whether or not this is related to the human infection theory, this might also be worth a mention.

Sophia David

Reviewer #4 (Remarks to the Author):

In this manuscript, Wee et al. describe a population genomic study of *L. pneumophila* clinical and environmental isolates that they performed to investigate genotypes associated with human disease. A genome-wide association study allowed them to identify the gene *lag-1*, which encodes an O-acetyltransferase that modifies LPS, as being strongly associated with clinical isolates. This gene appears to have been distributed across all major phylogenetic clades of *L. pneumophila* by frequent recent recombination events. Through the use of strains that vary in functional Lag-1 expression, as well as the introduction of *lag-1* variants encoded on an expression plasmid into a strain encoding a non-functional *lag-1* gene, the studies indicate that *lag-1* confers resistance to complement-mediated killing in human serum by inhibiting the deposit of complement proteins on the bacterial surface, inhibiting phagocytosis by human neutrophils, and promoting survival in a mouse model. Overall, this is a really interesting study and provides some potential new insight into the role of *lag-1* in allowing *L. pneumophila* to resist complement-mediated killing. Below are some comments and suggestions critical for strengthening the conclusions and the manuscript so that it is suitable for publication.

Major comments:

1. In Figure 6B, the authors' data indicate that there is an increase in strain 4451 CFUs in the factor B-depleted serum conditions to similar CFUs as observed for C1q-depleted serum. There also seems to be an increase in strain 4681 CFUs in factor B-depleted serum, although not to similar levels. Is this due to a biological difference in the role of the alternative pathway in complement killing of these two *L. pneumophila* strains, or a technical issue in the experiment shown? Have the authors tried examining the role of factor B or C1q in complement-mediated killing of other *L. pneumophila* strains?
2. For Figure 7A and C, as well as Supplementary figure 8, the authors should show representative flow cytometry plots with the gate for DsRED or FITC drawn so that it can be better ascertained by the reader how the MFI was determined.
3. For Figure 7, the authors conclude that the difference in neutrophil-mediated phagocytosis is due to differences in complement binding. If that is the case, is neutrophil-mediated phagocytosis of the different *L. pneumophila* strains that express or do not express *lag-1* similar in the absence of human serum?
4. For Figure 7E, it is recommended that the authors conduct earlier timepoints (24, 48, and/or 72 hrs) in addition to the 96 hour timepoint to obtain a better understanding of how *lag-1* expression affects the kinetics of infection. In addition, to test if the complement pathway accounts for increased control of the empty vector-strain, the authors should consider examining whether mice lacking the classical complement pathway would have similar CFU levels of the two strains following infection.
5. The paper is missing a discussion section.

Minor comments:

1. In figure 7a, it would be helpful to have a color key so that it is easier for the reader to know what color corresponds to what *lag-1* variant.
2. For Figure 7B and D, the authors should quantify the microscopy data in addition to showing representative images

REVIEWER COMMENTS

Reviewer #1 (Remarks to the Author):

"Population analysis of Legionella pneumophila reveals the basis for resistance to complement-mediated killing" by Wee and Alves et al.

In this work, the authors aim to answer why some *L. pneumophila* are associated with clinical disease in humans while others are not, a feature which to only some degree is dictated by clonal association. They do this by gathering a diverse collection of clinical and environmental samples and perform a genome-wide association study to investigate elements that are enriched in the clinical samples, and additionally validate their findings with functional assays.

I was quite impressed with the "soup to nuts"-approach methodology here - The authors go from collecting samples, whole-genome sequencing, in silico analysis, to wet lab transformation, in vitro functional validation of their findings and even in vivo in mice. The standard of evidence is really high here and it is obvious that the paper actually delivers on the promise from the title, i.e. the basis, at least partly, for complement resistance. I wish more functional studies followed this example! I am not competent to evaluate the wet lab work described here, but the bioinformatic work is performed according to a gold standard and the figures are useful and to the point. In short, it's a solid paper without any significant technical or logical shortcomings.

We are grateful for the positive comments.

I would only like to raise two minor issues related to reproducibility:

- I don't think the results could be reliably reproduced since Treemmer has a random component, and as far as I could tell there is no indication of which isolates were selected in each pruning step.

We thank the reviewer for raising this point. We agree that the precise set of isolates that is selected after applying Treemmer may vary between runs, but the role of the script is to reduce redundancy and the outputs of the GWAS analysis should not vary substantially. This has been tested by carrying out multiple subsampling runs followed by GWAS analysis which led to the same GWAS hits identified. We now present the results of 3 independent runs in a new panel in Supplementary Figure 2 (Supplementary Figure 2b) (lines 165-167).

- Most of the cited software are lacking version numbers.

Version numbers have now been added.

Reviewer #2 (Remarks to the Author):

The article « Population analysis of Legionella pneumophila reveals the basis for resistance to complement-mediated killing » by Wee and colleagues report a population genomics analyses based on 902 Legionella pneumophila strains with the aim to identify factors important for human infection. Genome sequencing of 397 Scottish *L. pneumophila* strains isolated from clinical cases or the environment was undertaken for this study, the remaining strains had been retrieved from public databases. Based on a GWAS analyses the authors report that the cluster coding for the LPS biosynthesis of SG-1 was highly enriched, and one gene of this cluster, lag-1 showed the strongest

association with clinical isolates. The authors further showed that recombination led to the dissemination of different alleles of lag-1 but that three are dominant in the population. Based on an early study of nosocomial *L. pneumophila* that reported that this strain survived complement killing compared to an environmental strain, the authors set out to analyze this phenotype and to investigate whether it is conferred by the presence of lag-1 in the genome. Using human blood from different donors, they tested representatives of the three different lag-1 carrying strains and strains lacking lag-1 with respect to serum killing. The authors show that lag-1 coding strains had higher resistance to killing in human plasma and that naturally occurring lag-1 inactivating mutants were less resistant. Using different inhibitors of the complement pathway they then further showed that lag-1 confers resistance to complement mediated killing. Finally, using human neutrophils and a mouse model of pulmonary infection, the authors report that lag-1 confers resistance to phagocytosis by human neutrophils and enhances virulence in pulmonary mouse infection.

The authors present a large body of work that seems to be well analyzed and interpreted and that is nicely illustrated with very comprehensive figures. The article is well presented and provides new data for a better understanding of the higher association of specific clones of *L. pneumophila* with human disease, however, some points need further clarification.

We thank the reviewer for the positive comments.

General comments:

The authors divide the strains they analyze into clinical and environmental strains and find a high association of the lag-1 gene with clinical strains. However, although clinical strains can be clearly defined as they caused disease, this is not so clear for “environmental” strains in particular when relying on strain information obtained from Public database deposited strains. Furthermore, several of the environmental strains have been collected when searching for outbreak sources and are labelled environmental. How the authors can ascertain that the strain definitions “clinical” and “environmental” are well assigned, as this definition is the basis for their GWAS analyses?

We thank the reviewer for making this important point. As pointed out, there will be some isolates that are labelled as ‘environmental’ that were sampled as part of an outbreak investigation and may in some cases be closely-related to clinical isolates with similar pathogenic potential. Accordingly, this approach will have led to some overlap between the environmental/clinical groupings that will reduce the overall power to identify pathogenic traits. In spite of this, the identification of *lag-1* was well supported using multiple GWAS approaches in the current study but future studies examining large groups of isolates with more strictly defined phenotypes will likely reveal additional pathogenic traits. We have now included text to this effect in the Discussion (lines 105-108).

I am confused with respect to the evolutionary history of lag-1. *L. pneumophila* are accidental, human pathogens, they are not transmitted from humans to humans, evolution for being “more virulent” does not take place in humans as the selection pressure is the environment, protozoa. Thus, how an evolution for enhanced resistance to serum killing should take place? There seems even be a negative selection for lag-1 in the environment as stated also by the authors (lines 268-269). How the authors explain this selection of specific lag-1 alleles for being associated to human infection?

We thank the reviewer for raising this interesting point. Indeed, the generally accepted principle is that *L. pneumophila* cannot usually be transmitted between humans and that human infections arise through acquisition from an environmental source. In which case, *lag-1* likely confers a selective

advantage in the environment, possibly in relation to interactions with its native host, free-living amoeba. It is feasible that *lag-1* also happens to provide an advantage in the human host (co-selection) and indeed many virulence traits of *L. pneumophila* are important for both human and amoebal host interactions. An alternative (but as yet unproven) possibility is (as proposed by reviewer 3), *L. pneumophila* can be asymptotically transmitted among human populations in which case, there may be a selective advantage for *lag-1* based on survival on the human host. We have now added some text to the Discussion that raises this interesting hypothesis (lines 224-232 and lines 407-416).

Why many strains carry non-functional pseudogenes of *lag-1*? (lines 199-200) but at the same time the authors state that there is a strong selection pressure for *lag-1* function? (lines 201-203). Could the authors speculate what the role of *lag-1* is in the environment?

Indeed, we report here an enrichment of *lag-1* loss of function mutations, particularly associated with a lineage endemic in a hospital environment. We speculate that the hospital niche occupied provides a negative selection perhaps due to the absence of its preferred amoebal host species. Consistent with this, in a study examining *L. pneumophila* survival in ultra-pure water, mAb 3/1-negative strains had a relative growth advantage (Schrammel, B. *et al. Water Res* **141**, 417-427, doi:10.1016/j.watres.2018.04.027 (2018))

While the selection for *lag-1* in the environment is unclear, multiple studies have provided evidence for different roles including adherence to and invasion of its amoebal host (Palusinska-Szys et al, *Front Microbiol* **10**, 2890, doi:10.3389/fmicb.2019.02890 (2019)), or increased hydrophobicity promoting survival inside vacuolar compartments (Fernandez-Moreira et al *Infect Immun* **74**, 3285-3295, doi:10.1128/IAI.01382-05 (2006)). Each of these possibilities has been discussed in the text (lines 221-224 and lines 407-411)

I am puzzled by the describing of the assays defining serum resistance of the different strains. According to the M&M section bacteria were grown on plate, then precultured, new medium was inoculated and then a culture harvested at OD₆₀₀ = 0.8 was used for assessing resistance to killing by human plasma or phagocytosis by neutrophils. *L. pneumophila* is known to have a biphasic life cycle and only bacteria grown to post exponential phase (e.g. OD 3.5 depending on the conditions) are known to be virulent and infectious. Thus bacteria at OD=0.8 are far from being in the PE phase but are in exponential phase. Could the authors explain? The authors should test PE phase bacteria if they give the same result.

We thank the reviewer for raising this point. As requested, we have now tested the susceptibility to serum killing of isolates at post-exponential phase (PE). Since the manuscript was submitted, the original supplement used to culture *L. pneumophila*, (Oxoid BCYE Growth Supplement SR0110) was discontinued and an alternative supplement made available (BCYE Growth Supplement LS0053, EO Labs). Accordingly, we carried out growth curves of *L. pneumophila* strains in the new media prior to determining the appropriate time-point for harvesting in PE phase (OD₆₀₀~1.4; Supplementary Figure 10). We tested serum resistance for *L. pneumophila* strains grown to PE phase that are representative of the three *lag-1* variants and three *lag-1* negative strains. In each case, the result was consistent with our previous data indicating resistance to killing associated with *lag-1*. These data are included in a new supplementary figure (Supplementary Figure 10). In addition, we have now included information in the Methods section about the new supplement and media used to generate the new data presented (Lines 431-436).

Minor comments:

Line 47, reference 1 is a reference from 2011, there are several newer ones that should be added or cited in place. e.g. Boamah et al., Front cell Infect Microbiol 2017; Mondino et al. Annu Rev Pathol. 2020

We have now included the references suggested by the reviewer.

Line 57, please add reference 11 here

The reference has now been added.

Line 119 please add a reference for these data

The statement on line 119 (“These data indicate that although only 5 STs are responsible for almost half of all infections, the other half of human infection potential comes from diverse genetic backgrounds distributed across the species.”) describes findings from the data in the current study.

Line 132 it reads... “The factors that contribute to the enhanced human pathogenic capacity of some *L. pneumophila* clones are unknown“..... it could also be enhanced infectivity or transmissibility

We thank the reviewer for pointing the possible effect of enhanced transmissibility as a factor contributing to the association to human infections of some *L. pneumophila* clones. We have changed the sentence to: “The factors that contribute to the enhanced human infectivity or transmissibility of some *L. pneumophila* clones are unknown” (lines 140-141).

Lines 148 – 153 the genes that were determined by GWAS analyses to be highly associated to human disease clones were identified as the LPS gene cluster of Sg1... please add the reference Cazalet et al. Genome Research 2010, and discuss that the LPS cluster has also been identified in this comparative genome analyses using microarrays, as associated to human disease.

We thank the reviewer for pointing this out. We introduced the reference and text as requested (lines 230-231).

Lines 182 -... where the different alleles of lag-1 are described I was missing an alignment of the three major alleles to appreciate the differences. Please add a definition on what basis the different alleles were defined.

We appreciate the comment highlighting the lack of clarity in the difference between the three major variants. We have now included information about the nucleotide sequence identity divergence between the three variants in the results and discussion (lines 215-218) and modified Supplementary Table 3 for improved clarity.

Supplementary Table 3a: please add a Table legend to explain what the different parts of the Table show.

We have now included a more descriptive Table legend.

Line 211, please add reference David et al., PLoS Genet. 2017 Jun 26;13(6):e1006855. Here

recombination among LPS gene clusters has already been described and Cazalet et al 2010, who described HGT of the LPS cluster among strains.

The references have now been included.

Line 227-230 it reads" A comparison of re-assortment rates across major phylogenetic lineages showed no significant differences between LPS and non-LPS genes across the *L. pneumophila* genome ($p=0.6275$), suggesting that recombination is active on a genome-wide scale " This has already been shown previously in David et al., PLoS Genet. 2017 Jun 26;13(6):e1006855 and Gomez-Valero et al., BMC Genomics 2011 Nov 1;12:536. please mention and cite these references.

We thank the reviewer for pointing these missing references. We have now include them in the text. (lines 250-251)

Reviewer #3 (Remarks to the Author):

The study by Wee et al. investigates the genetic basis of pathogenicity in the bacterial pathogen, *Legionella pneumophila*, using combined genomic and experimental approaches. This is an important area of research since our lack of understanding to date of why some *L. pneumophila* strains are over-represented among disease cases has constrained our ability to improve various aspects of disease control. New knowledge has the potential to enable more targeted environmental risk assessment and therapeutic strategies, and will also shed light on how this environmental bacterium has evolved to become a cause of serious disease.

Using a genome-wide association study (GWAS) with a large collection of clinical and environmental isolates, the authors identify a single gene, *lag-1*, involved in LPS modification, to be strongly associated with clinical isolates. They show that *lag-1* has been distributed horizontally across all major clades of the species by recent recombination events. Using experimental approaches, they demonstrate the mechanism by which *lag-1* increases pathogenicity - by inhibiting complement-mediated killing and reducing phagocytosis by human neutrophils. Finally, they show that *lag-1* also promotes survival in a murine model of pulmonary legionellosis. Overall this is a beautiful story with an exciting finding, and provides an important advance in our understanding of *L. pneumophila* evolution and pathogenicity. The genomic analyses (for which I have been specifically asked to comment) have been performed to the highest standard, and generally the manuscript is clearly presented and contains detailed methodology. As such, I only have minor comments and suggestions listed below, as well as a more general discussion point.

We thank the reviewer for the supportive comments.

Lines 97-98. Are the clinical isolates from Legionnaires' disease patients (presumably), or are any from Pontiac fever patients? This seems relevant given the focus on pathogenicity, but may not be obvious to a non-*Legionella* audience.

We thank the reviewer for this query. We have now clarified in the text that we presume Pontiac fever isolates are very limited or absent due to low hospitalisation rates for these mild infections (lines 98-100).

Lines 98-99. If some of the environmental isolates are outbreak-related, it might be worth a comment on how this would affect the GWAS?

We thank the reviewer for making this important point. As suggested, there may be some isolates that are labelled as 'environmental' that were sampled as part of an outbreak investigation and may in some cases be related to clinical isolates and presumably have pathogenic potential. This approach will have led to some overlap between the environmental/clinical groupings that will reduce the overall power to identify pathogenic traits. The identification of *lag-1* was well supported using multiple GWAS approaches in the current study but future studies examining large groups of isolates with more strictly defined phenotypes will likely reveal additional pathogenic traits.

We have now included text to this effect in the Discussion (lines 105-108).

Lines 124-126. This assumes that the *L. pneumophila* obtained from cooling towers and other artificial environments are representative of environmental *L. pneumophila*. The wording could be tweaked to something like...

"Notably, 35% (81 of 231) of environmental isolates belong to one of the five clinically dominant STs suggesting that at least a third of all environmental *L. pneumophila* *from the sampled sources* in Scotland have human pathogenic potential."

We agree with the reviewer and have changed the sentence as suggested (lines 133-134).

Line 145. Would it be useful to mention here that the Philadelphia genome is sg-1?

We have now included the information about the serogroup.

Lines 148-152. Are these genes with significant k-mer hits, including *lag-1*, specific to sg-1?

We thank the reviewer for this query. While some of the k-mer hits are in fact specific for Sg-1 isolates (eg for *lag-1*), not all are exclusive to this serogroup.

Lines 160-161.

"...this approach also indicated that the LPS biosynthesis genes were enriched among clinical isolates..."

Would it be clearer to say "sg-1 LPS biosynthesis genes"? It becomes clear that you mean that the sg-1 LPS genes are enriched (rather than LPS genes more generally) in both the SEER and Scoary analyses, but I think this could be clarified more here and in the preceding text (this relates to the previous two comments).

We thank the reviewer for pointing this out, we have changed the sentence to include the information about the serogroup.

Lines 131-180. Do you know why other LPS genes are coming up in the GWAS? Is it because these are sg-1 genes that are more likely to be found with *lag-1*, or could these be important independent of *lag-1*?

We hypothesise that the other LPS genes are also highlighted in the GWAS due to linkage disequilibrium. In general, there are fewer significant *k*-mers mapping to genes further away from *lag-1*. This hypothesis is also reinforced by the observation that the gene phylogeny of the LPS genes

closest to *lag-1* are more similar to the gene phylogeny of *lag-1*.

Lines 131-180. Having identified *lag-1* through statistical methods, it could be useful to then provide the number of clinical vs environmental genomes that *lag-1* is found in. It might also be striking to see the clinical/environmental and *lag-1* presence/absence rings on the same tree, but appreciate too that this might look more messy.

We thank the reviewer for this suggestion. We have now included the proportion clinical and environmental isolates that have *lag-1* and a new supplementary figure that shows the distribution of all *lag-1* variants on the phylogeny is included (Supplementary Figure 3) (lines 179-180).

Lines 200-203. It's fascinating that the *lag-1* gene appears to have spread recently over different clades of *L. pneumophila*. I wonder whether the evidence for recent horizontal spread could be improved by putting the amount of *lag-1* variation into context, i.e. how much lower is the allelic variation in *lag-1* than in other core genes? Also, how long is the *lag-1* gene? (I would be more convinced that identical sequences indicate recent horizontal spread if the gene is not very small.) It would also be interesting to know how different the three major variants are from each other.

We thank the reviewer for this suggestion. The *lag-1* gene is 1,017 bp long, similar to the average length of a gene in *L. pneumophila* (mean: 1024 bp). The nucleotide sequence identity of the three *lag-1* allelic variants is from 89% - 95% , and we have now added this information in the results and discussion section (Lines 215-218)

Lines 446. "not more less than 200 contigs of more than 1Kb."
I think this needs correction?

We thank the reviewer for pointing this error. The sentence has been replaced with "fewer than 200 contigs of 1 kb or longer" (line 492)

Lines 457. github.com/bawee/Treemmer doesn't seem to be available.

This issue has been fixed and the repository now updated.

Figures 1 and 3. The tree scale should be described in the legend.

The tree scale was added to relevant Figure legends as suggested (line 888-889, line 908, line 995 and line 1013).

Table S3. Do the values in the table refer to % nucleotide divergence?

Yes, the table values refer to % of nucleotide divergence. To improve clarity we have changed the Table legend to Table S3. A distance matrix indicating the percentage nucleotide identity divergence between representative allelic variants of *lag-1* sequences identified in the dataset" and we have edited the Table column headings.

Is the discovery of a gene that enhances the ability of *L. pneumophila* to infect humans (and mice) and which has spread rapidly over recent years not a very intriguing finding, also in light of the recent expansion of major disease-associated clades? Why would this be the case? Previous work has shown that multiple different *Lp* strains have spread rapidly around the world and found

themselves in lots of different man-made environments (as evidenced from the diversity of places that people become infected). The repeated, recent acquisition of lag-1 that you show has occurred in many of these different lineages likely suggests one common selection pressure. Perhaps that selection pressure is due to one common amoebal species that is present in all these different man-made environments, but couldn't it also be possible that the one shared feature is human exposure and these strains are adapting to (mostly asymptomatic/mild) human infection? That also avoids the need for additional explanation of how an adaptive trait relating to amoebae survival/infection (or other hosts/conditions) also co-incidentally results in increased virulence in humans. I appreciate that the authors say that the selective advantage of lag-1 is "unclear" and thus leave any other speculation on this matter to their own discretion!

We agree with the reviewer that human transmission of *L. pneumophila* is an attractive hypothesis and that our data may be consistent with such a scenario, though strong evidence is currently lacking. We thank the reviewer for raising this interesting point and we have now added some text to the Discussion that speculates on this possibility (lines 224-232 and lines 407-416).

What's also striking to me is that the phylogenetic distribution of lag-1 suggests that its acquisition may have driven some of the very recent clonal expansions (e.g. most clear for ST47 and ST42), likely in combination with other factors. Whether or not this is related to the human infection theory, this might also be worth a mention.

We have now added a comment to this effect (lines 224-228).

Sophia David

Reviewer #4 (Remarks to the Author):

In this manuscript, Wee et al. describe a population genomic study of *L. pneumophila* clinical and environmental isolates that they performed to investigate genotypes associated with human disease. A genome-wide association study allowed them to identify the gene lag-1, which encodes an O-acetyltransferase that modifies LPS, as being strongly associated with clinical isolates. This gene appears to have been distributed across all major phylogenetic clades of *L. pneumophila* by frequent recent recombination events. Through the use of strains that vary in functional Lag-1 expression, as well as the introduction of lag-1 variants encoded on an expression plasmid into a strain encoding a non-functional lag-1 gene, the studies indicate that lag-1 confers resistance to complement-mediated killing in human serum by inhibiting the deposit of complement proteins on the bacterial surface, inhibiting phagocytosis by human neutrophils, and promoting survival in a mouse model. Overall, this is a really interesting study and provides some potential new insight into the role of lag-1 in allowing *L. pneumophila* to resist complement-mediated killing. Below are some comments and suggestions critical for strengthening the conclusions and the manuscript so that it is suitable for publication.

We are grateful for the positive comments.

Major comments:

1. In Figure 6B, the authors' data indicate that there is an increase in strain 4451 CFUs in the factor B-depleted serum conditions to similar CFUs as observed for C1q-depleted serum. There also seems to be an increase in strain 4681 CFUs in factor B-depleted serum, although not to similar levels. Is this due to a biological difference in the role of the alternative pathway in complement killing of these two *L. pneumophila* strains, or a technical issue in the experiment shown? Have the authors tried examining the role of factor B or C1q in complement-mediated killing of other *L. pneumophila* strains?

We thank the reviewer for highlighting this difference between the 2 strains with regard to the level of the effect of factor B depletion. We do not believe that it is a technical issue since each point in figure 6B is from an independent experiment and there is a clear difference between the values for the strains 4451 and 4681, consistent with strain-dependent variation. We have not tried examining the effect in additional strains but this would be an interesting idea for future studies. We have now included some text relating to the strain-dependent variation in the role of the alternative pathway (lines 341-343).

2. For Figure 7A and C, as well as Supplementary figure 8, the authors should show representative flow cytometry plots with the gate for DsRED or FITC drawn so that it can be better ascertained by the reader how the MFI was determined.

The MFI for the neutrophil experiments was calculated on the total single cell gate, as described in the legend of the supplementary figure 8 (now supplementary figure 9, line 1063-1064) and Methods section (line 597-599), and the flow cytometry plots in the supplementary figure 8 (now supplementary figure 9) indicate the gate from where the MFI was determined.

3. For Figure 7, the authors conclude that the difference in neutrophil-mediated phagocytosis is due to differences in complement binding. If that is the case, is neutrophil-mediated phagocytosis of the different *L. pneumophila* strains that express or do not express lag-1 similar in the absence of human serum?

Thank you for raising this query. We have demonstrated that there is no internalization of bacteria by the neutrophils in the absence of human serum (now supplementary figure 9b) and the requirement of serum for *L. pneumophila* phagocytosis by human neutrophils is stated in lines 378-380.

4. For Figure 7E, it is recommended that the authors conduct earlier timepoints (24, 48, and/or 72 hrs) in addition to the 96 hour timepoint to obtain a better understanding of how lag-1 expression affects the kinetics of infection. In addition, to test if the complement pathway accounts for increased control of the empty vector-strain, the authors should consider examining whether mice lacking the classical complement pathway would have similar CFU levels of the two strains following infection.

We agree with the reviewer that it would be interesting to examine early time-points and to use mouse lines that lack the complement system but respectfully feel that these would be beyond the scope of the current study.

5. The paper is missing a discussion section.

We have employed a structure that includes a combined Results and Discussion section.

Minor comments:

1. In figure 7a, it would be helpful to have a color key so that it is easier for the reader to know what color corresponds to what lag-1 variant.

We thank the reviewer for this helpful suggestion and have now added a colour key to Figure 7a.

2. For Figure 7B and D, the authors should quantify the microscopy data in addition to showing representative images

The microscopy images are designed to provide a qualitative indication of neutrophil uptake to complement the flow cytometric analysis. Accordingly, we suggest that quantification of the microscopy data would not add any value to the more accurate quantification provided by the flow cytometry. However, if the reviewer feels strongly that it should be included, we can do so.

We thank the reviewers for their constructive comments which have helped to improve the manuscript markedly.

REVIEWERS' COMMENTS

Reviewer #2 (Remarks to the Author):

The authors have answered my questions and concerns and the revised manuscript is improved. I have no further requests.

The only point that remains and will remain unanswered is the evolutionary pressure leading to the selection of lag-1 for being advantageous in human infection and in serum resistance, a trait not needed in the environmental life of Legionella. The authors propose that *L. pneumophila* may be asymptotically transmitted among human populations and this might lead to this selection pressure... however no proof for this suggestion is available. To my knowledge all studies analyzing a possible asymptomatic carriage of Legionella failed to proof such a stage, thus this suggestion is very hypothetical.

Reviewer #3 (Remarks to the Author):

The authors have done an excellent job in addressing the reviewers' queries.

I have only one comment, relating to the sentence in lines 412-415.

"Alternatively, David et al previously proposed that the recent expansion of *L. pneumophila* in man-made water systems and global spread of selected clones is consistent with possible human-to-human dissemination associated with asymptomatic or mild infections."

In this paper, we suggested transmission may be human-human or human-environment, so this point could be clarified here for accuracy. We also did not speculate on the nature of the infections (e.g. asymptomatic, mild, etc). This last comment also relates to the new sentence in lines 225-227.

Reviewer #4 (Remarks to the Author):

The authors have satisfactorily addressed the reviewers' concerns.

RESPONSE TO REVIEWERS' COMMENTS

Reviewer #2 (Remarks to the Author):

The authors have answered my questions and concerns and the revised manuscript is improved. I have no further requests.

Authors' Response: We appreciate the reviewer's constructive comments throughout the reviewing process.

The only point that remains and will remain unanswered is the evolutionary pressure leading to the selection of lag-1 for being advantageous in human infection and in serum resistance, a trait not needed in the environmental life of Legionella. The authors propose that *L. pneumophila* may be asymptotically transmitted among human populations and this might lead to this selection pressure... however no proof for this suggestion is available. To my knowledge all studies analyzing a possible asymptomatic carriage of Legionella failed to proof such a stage, thus this suggestion is very hypothetical.

Authors' Response: We agree with the reviewer that the transmission from asymptomatic carriers is hypothetical and speculative. We have edited the sentences in lines 412-415 and 225-227 replacing the reference to asymptomatic human transmission to "between humans or from humans to the environment" as previously proposed by David et al.

Reviewer #3 (Remarks to the Author):

The authors have done an excellent job in addressing the reviewers' queries.

Authors' Response: We thank the reviewer for the constructive comments provided during the review process.

I have only one comment, relating to the sentence in lines 412-415.

"Alternatively, David et al previously proposed that the recent expansion of *L. pneumophila* in man-made water systems and global spread of selected clones is consistent with possible human-to-human dissemination associated with asymptomatic or mild infections."

In this paper, we suggested transmission may be human-human or human-environment, so this point could be clarified here for accuracy. We also did not speculate on the nature of the infections (e.g. asymptomatic, mild, etc). This last comment also relates to the new sentence in lines 225-227.

Authors' Response: We have now edited the sentences in lines 412-415 and 225-227 replacing the reference to transmission by asymptomatic carriers to highlight the possibility of either human-to-human or human-to-environmental dissemination.

Reviewer #4 (Remarks to the Author):

The authors have satisfactorily addressed the reviewers' concerns.

Authors' Response: We are grateful for the reviewer's comments throughout the review process.